# H3K9me3 is required for inheritance of small RNAs that target a unique subset of newly evolved genes

**Itamar Lev\*, Hila Gingold, Oded Rechavi\***

Department of Neurobiology, Wise Faculty of Life Sciences & Sagol School of Neuroscience, Tel Aviv University, Tel Aviv, Israel

**Abstract** In *Caenorhabditis elegans*, RNA interference (RNAi) responses can transmit across generations via small RNAs. RNAi inheritance is associated with Histone-3-Lysine-9 tri-methylation (H3K9me3) of the targeted genes. In other organisms, maintenance of silencing requires a feed-forward loop between H3K9me3 and small RNAs. Here, we show that in *C. elegans* not only is H3K9me3 unnecessary for inheritance, the modification's function depends on the identity of the RNAi-targeted gene. We found an asymmetry in the requirement for H3K9me3 and the main worm H3K9me3 methyltransferases, SET-25 and SET-32. Both methyltransferases promote heritable silencing of the foreign gene *gfp*, but are dispensable for silencing of the endogenous gene *oma-1*. Genome-wide examination of heritable endogenous small interfering RNAs (endo-siRNAs) revealed that endo-siRNAs that depend on SET-25 and SET-32 target newly acquired and highly H3K9me3 marked genes. Thus, 'repressive' chromatin marks could be important specifically for heritable silencing of genes which are flagged as 'foreign', such as *gfp*.

**Editorial note:** This article has been through an editorial process in which the authors decide how to respond to the issues raised during peer review. The Reviewing Editor's assessment is that all the issues have been addressed (see decision letter).

DOI: https://doi.org/10.7554/eLife.40448.001

**\*For correspondence:**
itamai.et@gmail.com (IL);
odedrechavi@gmail.com (OR)

**Competing interests:** The authors declare that no competing interests exist.

## Introduction

RNA interference (RNAi) responses are inherited in *Caenorhabditis elegans* nematodes across generations via heritable small RNAs (*Alcazar et al., 2008*; *Buckley et al., 2012*; *Vastenhouw et al., 2006*). In worms, exposure to a number of environmental challenges, such as viral infection (*Gammon et al., 2017*; *Rechavi et al., 2011*), starvation (*Rechavi et al., 2014*), heat (*Klosin et al., 2017*), and growth in liquid (*Lev et al., 2018*) induces heritable physiological responses that persist for multiple generations. Inheritance of such transmitted information was linked to inheritance of small RNAs and chromatin modifications, and hypothesized to protect and prepare the progeny for the environmental challenges that the ancestors met.

By base-pairing with complementary mRNA sequences, small RNAs in *C. elegans* control the expression of thousands of genes, and protect the genome from foreign elements (*Luteijn and Ketting, 2013*; *Malone and Hannon, 2009*). Via recruitment of RNA-binding proteins, small interfering RNAs (siRNAs) can induce gene silencing also by inhibiting transcription (*Castel and Martienssen, 2013*).

Small RNA-mediated transcription inhibition involves modification of histones, however the exact role that histone marks play in inheritance of RNAi and small RNA synthesis is still not entirely clear (*Rechavi and Lev, 2017*). In *C. elegans* small RNAs that enter the nucleus were shown to inhibit the elongation phase of Pol II (*Guang et al., 2010*); In addition, nuclear small RNAs are thought to recruit histone modifiers to the target's chromatin, resulting in deposition of histone marks such as

histone H3K9-tri methylation (H3K9me3) and H3K27me3 (*Gu et al., 2012*; *Lev et al., 2017*; *Mao et al., 2015*).

The interactions between small RNAs and repressive chromatin marks are reciprocal: in *Arabidopsis thaliana* (*Holoch and Moazed, 2015*; *Molnar et al., 2010*) and *Schizosaccharomyces pombe* (*Moazed et al., 2006*; *Verdel et al., 2004*; *Hall et al., 2002*) small RNAs and repressive histone marks form a self-reinforcing feed-forward loop, where nuclear small RNAs induce deposition of repressive histone marks, and in turn the repressive chromatin marks recruit the small RNA machinery to synthesize additional small RNAs. Whether a similar feedback operates in worms and other organisms, is still under investigation. In *Neurospora crassa*, transgene-induced small RNAs work independently of H3K9me3 (*Chicas et al., 2005*). In *C. elegans*, it was previously suggested that H3K9me is required for RNAi inheritance (*Shirayama et al., 2012*). However, studies from different groups have shown that the situation is more complex, and that H3K9me could be dispensable, and can even suppress heritable silencing of some targets (*Kalinava et al., 2017*; *Lev et al., 2017*; *Minkina and Hunter, 2017*).

In *C. elegans* H3K9me is considered to depend mainly on the methyltransferases MET-2, SET-25, and SET-32 (*Kalinava et al., 2017*; *Spracklin et al., 2017*; *Towbin et al., 2012*). H3K9 methylation by MET-2 and SET-25 occurs in a step-wise fashion – after MET-2 deposits the first two methyl groups (H3K9me1/2), SET-25 can add the third methyl group (me3) (*Towbin et al., 2012*). In the germline, however, SET-25 is capable of tri-methylating H3K9 in a MET-2-independent manner (*Bessler et al., 2010*; *Towbin et al., 2012*). SET-32-dependent H3K9me3 is at least in part independent of the activity of SET-25 or MET-2 (*Kalinava et al., 2017*).

To study the roles of H3K9me3 in the maintenance of heritable small RNAs, we examined the inheritance of small RNAs in mutants defective in these histone methyltransferases. Although H3K9me3 was thought to be required for heritable RNAi (*Ashe et al., 2012*; *Gu et al., 2012*), we found the heritable RNAi-responses are greatly potentiated in *met-2* mutant background (*Lev et al., 2017*). Our data indicated that the enhanced strength of the RNAi responses in *met-2* mutants stems from a genome-wide massive loss of different endogenous small RNA (endo-siRNAs) species. In normal circumstances, these endo-siRNAs compete with exogenously derived siRNAs over shared biosynthesis components required for small RNA production or inheritance (*Lev et al., 2017*). In addition, we found that the accumulated sterility (or 'Mortal Germline', Mrt phenotype) of *met-2* mutants results from dysfunctional small RNA inheritance (*Lev et al., 2017*).

However, our previous results regarding the role of H3K9me1/2 (deposited by MET-2) did not rule out the possibility that H3K9me3 is yet required for efficient heritable silencing of *gfp* transgenes: We found that RNAi responses in *met-2* mutants nevertheless lead to marking of the target gene's histones with a heritable H3K9me3 modification. Further, a comparison of the H3K9me3 signal on the *gfp* locus in different mutants has shown that anti-*gfp* RNAi responses were strongly inherited only in genetic backgrounds where some H3K9me3 trace could be detected (i.e, in wild type, *met-2,* and *met-2;set-25* mutants). In *set-25* single mutants, where no statistically significant H3K9me3 footprint could be detected, anti-*gfp* RNAi was only weakly inherited. Previously, *set-25* mutants were reported to be deficient in heritable RNAi responses targeting different fluorescent transgenes (*Ashe et al., 2012*; *Lev et al., 2017*).

RNAi silencing of the endogenous *oma-1* gene is also inherited transgenerationally. In contrast to anti-*gfp* heritable RNAi responses, for which H3K9me3 is important, we detected an enhancement in the inheritance potency of anti-*oma-1* RNAi in *set-25* mutants (*Lev et al., 2017*). However, in that study we did not examine whether an H3K9me3 footprint was deposited on the endogenous gene *oma-1* in the *set-25* background (*Lev et al., 2017*). The publication of a recent paper (*Kalinava et al., 2017*) which described strong anti-*oma-1* RNAi inheritance in *met-2;set-25;set-32* triple mutants, despite the absence of a detectable H3K9me3 footprint, prompted us to re-examine the inheritance of anti-*gfp* RNAi in this triple mutants. We hypothesized that gene-specific characteristics lead to contrasting requirements for H3K9me3 and specific methyltransferases. In this manuscript, we describe an asymmetry in the requirement for H3K9me3 and specific methyltransferases for heritable RNAi responses aimed against the endogenous gene *oma-1* and the foreign gene *gfp*. These differences led us to perform a genome-wide analysis of H3K9 methyltransferase-dependent small RNAs, which revealed that the endo-siRNAs, which depend on H3K9me3 target newly acquired *C. elegans* genes that might be considered 'foreign', similarly to *gfp*.

## Results

Recently Kalinava et al. examined the heritable RNAi responses against *oma-1* also in a triple mutant, lacking the three main *C. elegans* H3K9 methyltransferases, SET-25, SET-32 and MET-2 (*Kalinava et al., 2017*). The authors reported that silencing of *oma-1* was independent of H3K9me3, as in these mutants RNAi responses raised against the *oma-1* gene were heritable despite the lack of an H3K9me3 trace (*Kalinava et al., 2017*).

We successfully replicated the results of Kalinava et al., and came to the same conclusion, that the *met-2;set-25;set-32* triple mutant worms inherit RNAi responses against the *oma-1* gene, also when we used a different assay for inheritance (*Figure 1A* and *Figure 1B*, upper panel). Unlike Kalinava et al., which used qPCR to score for downregulation of *oma-1* expression, we targeted a redundant, temperature-sensitive and dominant *oma-1* allele, that in the restrictive temperatures does not allow the development of embryos unless silenced (as previously described [*Alcazar et al., 2008*]). Upon shifting to 20 degrees, only worms that silence the *oma-1* gene in a heritable manner survive.

In parallel we discovered, surprisingly, that in contrast to anti-*oma-1* inheritance, heritable silencing of a *gfp* transgene was defective in the same triple mutants (*Figure 1C*, upper panel, p=0.0014, 2-way ANOVA). In addition, we also confirmed (*Spracklin et al., 2017*) that while *set-32* single mutants are deficient in inheriting RNAi responses raised against the *gfp* transgene (*Figure 1C*, lower panel, p=0.0026, 2-way ANOVA), they are capable (*Kalinava et al., 2017*) of inheriting responses raised against *oma-1* (*Figure 1B*, lower panel, p=0.8487, 2-way ANOVA). Previously we have shown that while *set-25* mutants are defective in inheritance of anti-*gfp* RNAi, weak inheritance responses can still be observed (*Lev et al., 2017*). Similarly, we were able to detect weak inheritance responses that last at least until the F3 generation also in *met-2;set-25;set-32* and *set-32* mutants (*Figure 1—figure supplement 1, p*-value < 0.0001 for *met-2;set-25;set-32* and *set-32* in the F3 generation, Two-way ANOVA). Together with our previous data, which showed that *set-25* is required for inheriting anti-*gfp* RNAi, but not anti-*oma-1* RNAi (*Lev et al., 2017*), these results suggested that heritable RNAi requires H3K9 methyltransferases in a gene-specific manner.

## The levels of RNAi-induced H3K9me3 do not explain the gene-specific requirements of methyltransferases for heritable RNAi

Histone methyltransferase mutants may affect RNAi-induced H3K9me3 levels in a gene-specific manner, thus leading to different inheritance dynamics for each gene. To test this possibility, we performed anti-H3K9me3 Chromatin Immunoprecipitation (ChIP) on F1 *met-2;set-25;set-32* triple mutant progeny, that were derived from parents exposed to anti-*oma-1* RNAi, anti-*gfp* RNAi, or untreated controls. Using qPCR we found, as was discovered before (*Kalinava et al., 2017*) that in *met-2;set-25;set-32* triple mutants the RNAi-induced H3K9me3 signal was significantly reduced (p-value=0.0007 and 0.0009, Two-way ANOVA, for *gfp* and *oma-1*, respectively). Importantly, this was true for both the *oma-1* and *gfp* loci (*Figure 2A*). Interestingly, in naive wild-type animals, that were not treated with RNAi, the levels of H3K9me3 on *gfp* were significantly higher than on *oma-1* (*Figure 2B*, p-value = 0.0039, Two-Way ANOVA) and an additional germline-expressed gene *dpy-28* (*Figure 2B*, p-value = 0.0176, student's t-test). We discuss the possible contribution of this RNAi-independent H3K9me3 signal below. Regardless, as no differences can be found in the RNAi-induced fold changes in H3K9me3 levels between *gfp* and *oma-1* (*Figure 2A*), the levels of *RNAi-induced* H3K9me3 cannot explain the gene-specific requirements of methyltransferases for heritable RNAi.

## SET-32 acts upstream to MET-2 and SET-25 to support RNAi inheritance

We previously found that in contrast to *set-25* single mutants, which are deficient in RNAi-induced heritable H3K9me3 methylation (*Lev et al., 2017*; *Mao et al., 2015*), *met-2;set-25* double mutants display a modest but robust H3K9me3 footprint following RNAi (*Kalinava et al., 2017*; *Lev et al., 2017*). We therefore hypothesized that in the *met-2* background, an additional, perhaps otherwise inactive H3K9 methyltransferase, is expressed or activated, compensating for the absence of SET-25, to allow efficient heritable RNAi responses (see *Figure 1—figure supplement 2* for summary). To test this hypothesis, we first examined whether *met-2;set-32* double mutants can inherit RNAi responses raised against *gfp*. If SET-32 and SET-25 compensate for each other and are redundant, then *met-2;set-32* double mutants are expected to strongly inherit RNAi responses, similar to *met-2;*

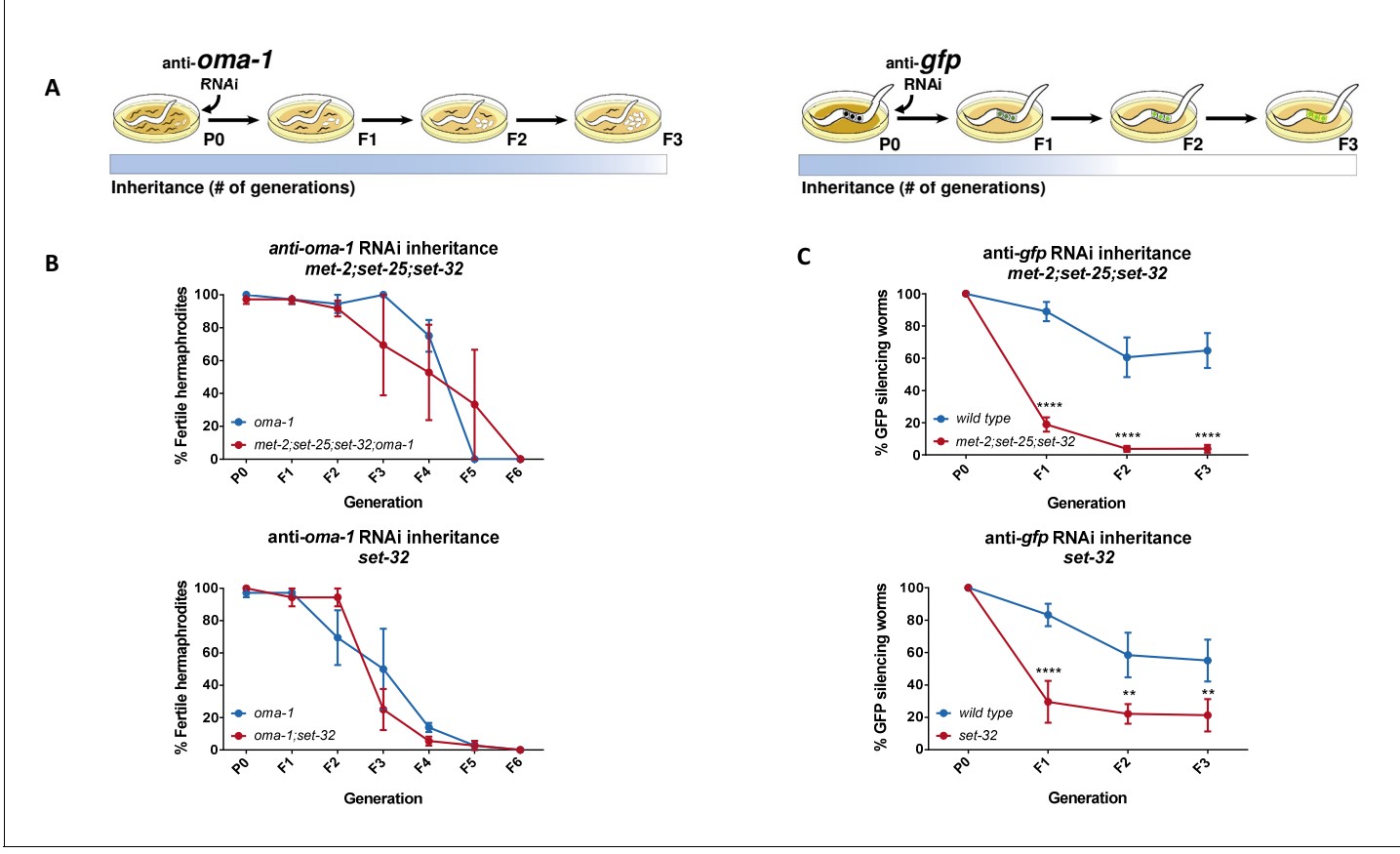

**Figure 1.** Heritable RNAi responses against the *oma-1* and *gfp* genes have different requirements for H3K9me3 methyltransferases. (**A**) Scheme depicting the different requirements for H3K9 methyltransferases in RNAi inheritance responses aimed at different genes. In the parental generation, worms are exposed to RNAi by growing on plates seeded with dsRNA-producing bacteria. Next, worms are transferred to plates seeded with control bacteria (that do not express dsRNA) to lay the eggs the next generation. (left) Only worms that inherit small RNAs that silence the temperature-sensitive dominant allele of *oma-1* can hatch. Heritable RNAi responses aimed against the endogenous *oma-1* gene do not require H3K9me3 methyltransferases. (right) Inheritance of anti-*gfp* small RNAs lead to heritable silencing of the *gfp* transgene (*Pmex-5::gfp::h2b transgene*). Heritable RNAi responses aimed against the foreign *gfp* gene strongly depends on H3K9me3 methyltransferases. (**B**) Inheritance of anti-*oma*-1 RNAi response in H3K9me3 methyltransferase mutants. The percentage of fertile worms per replicate and generation is presented (N = 12, three biological replicates). (upper panel) RNAi inheritance dynamics in *met-2;set-25;set-32;oma-1* mutants compared to *oma-1* mutants. (lower panel) RNAi inheritance dynamics in *set-32;oma-1* mutants compared to *oma-1* mutants. (**C**) Inheritance of anti-*gfp* RNAi response in H3K9me3 methyltransferase mutants. In each generation the percentage of worms silencing a germline expressed GFP transgene is presented (N > 60, five replicates). (upper panel) RNAi inheritance dynamics in *met-2;set-25;set-32* triple mutants. (lower panel) RNAi inheritance dynamics in *set-32* single mutants. Error bars represent standard error of mean. *p-value<0.05, **p-value<0.005, ***p-value<0.001, ****p-value<0.0001, Two-way ANOVA, Sidak's multiple comparisons test.

DOI: https://doi.org/10.7554/eLife.40448.002

The following figure supplements are available for figure 1:

**Figure supplement 1.** Weak anti-*gfp* heritable RNAi silencing of *gfp* in methyltransferases mutants.

DOI: https://doi.org/10.7554/eLife.40448.003

**Figure supplement 2.** SET-32 is required for the strong heritable RNAi-induced silencing of *gfp* in *met-2* mutants.

DOI: https://doi.org/10.7554/eLife.40448.004

**Figure supplement 3.** RNAi-induced silencing of *fog-2* or *sup-35* is not inherited transgenerationally.

DOI: https://doi.org/10.7554/eLife.40448.005

*set-25* double mutants (*Lev et al., 2017*). Our results show, that in contrast to *met-2;set-25* double mutants, *met-2;set-32* double mutants are defective in RNAi inheritance raised against *gfp*, since only a very weak response can be detected (*Figure 1—figure supplement 2A*). The potency of RNAi inheritance in *met-2;set-32* double mutants is comparable to that of *set-25* (*Lev et al., 2017*) and *set-32* single mutants, or *met-2;set-25;set-32* triple mutants (*Figure 1C*). These results suggest that SET-32 has a distinct role, and that it probably acts upstream to MET-2 and SET-25, in

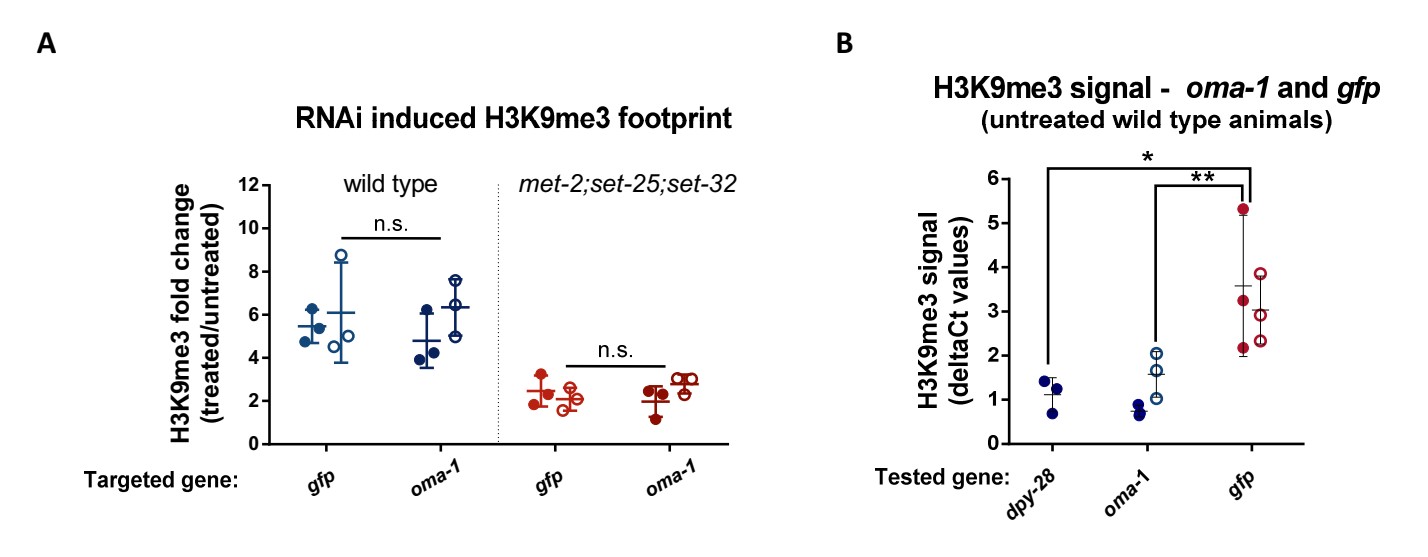

**Figure 2.** The fold change in RNAi-induced H3K9me3 on *oma-1* and *gfp* is comparable. (A) The RNAi-induced H3K9me3 footprint on the RNAi-targeted genes. The fold change in H3K9me3 levels in F1 progeny of animals exposed to RNAi versus untreated control animals. The H3K9me3 footprint levels were assessed using a qPCR quantification of ChIP experiments conducted with both wild type (left) and *met-2;set-25;set-32* mutants (right). Filled or empty circles represent qPCR data obtained using two different primer sets that span different parts of the examined locus. (B) H3K9me3 levels on the *gfp* and *oma-1* genes in naive untreated wild type animals. The deltaCt numbers used to obtain the fold change values were calculated using the *eft-3* gene as an endogenous control. The presented data were obtained from three biological replicates. The levels of *gfp* and *dpy-28* H3K9me3 signal in wild type animals are adapted from raw data from our previous publication (*Lev et al., 2017*). Two-way ANOVA, Sidak's multiple comparisons test. **p-value<0.005. Error bars represent standard deviations.
DOI: https://doi.org/10.7554/eLife.40448.006

promoting RNAi inheritance. This conclusion is also consistent with the recent observation that SET-32, in contrast to MET-2 and SET-25, has an essential role in the establishment of RNAi-mediated nuclear silencing (*Kalinava et al., 2018*).

### Unlike RNAi silencing of *oma-1*, silencing of *sup-35* and *fog-2* genes is not inherited transgenerationally

Currently, the only gene that serves to study heritable transgenerational (more than two generations) RNAi of endogenous genes is *oma-1*. Transgenerational RNAi inheritance requires the target gene to be expressed in the germline, and many germline genes are essential or do not have a phenotype that can be scored over many generations. The *oma-1* gene can serve as a tool for studying RNAi inheritance owing to the availability of a temperature-sensitive, dominant-lethal and redundant allele that can be rescued by RNAi (*Alcazar et al., 2008*). In search of other endogenous target genes whose heritable silencing could be studied, we examined the inheritance of RNAi against the non-essential germline genes *sup-35* and *fog-2*. SUP-35 is a maternally deposited toxin, expressed in the mother's germline, suppressed by PHA-1, a zygotically expressed anti-toxin (*Ben-David et al., 2017*). Consequently, temperature-sensitive *pha-1(e2123)* mutants develop when grown at 15 degrees but arrest their development when grown in restrictive temperatures, unless exposed to anti-*sup-35* RNAi. As previously described (*Ben-David et al., 2017*), RNAi silencing of *sup-35* allowes *pha-1* mutants to develop. However, we found this response was not inherited beyond the F1 generation (*Figure 1—figure supplement 3A*). Expression of the germline gene *fog-2* is required for hermaphrodite worms to produce sperm, but is dispensable for sperm production in males (*Schedl and Kimble, 1988*). Silencing of *fog-2* by RNAi lead to depletion of sperm (as evident by stacked oocytes), and the worms were unable to reproduce unless crossed with a male. While we found that this response was inherited to the F1 progeny, it was not inherited transgenerationally (*Figure 1—figure supplement 3B*). In conclusion, we could not find additional endogenous gene targets that can be transgenerationally silenced upon RNAi. Conveniently, many endo-siRNAs that target various

endogenous genes are inherited transgenerationally, and such inheritance can be studied using RNA sequencing.

## H3K9me3 methyltransferases are required for the biogenesis of a specific class of endo-siRNAs

Certain germline small RNAs have evolved to confer immunity against foreign genetic elements, while sparing endogenous genes (*Malone and Hannon, 2009*). The different requirements for particular methyltransferases and H3K9me3 for heritable silencing of *gfp* and *oma-1* may be connected to the fact that *gfp* is a 'foreign' gene, while *oma-1* is an endogenous gene. We previously found that exogenous siRNAs that target *gfp* are lost in *set-25* mutants, and hypothesized that endo-siRNAs that target other 'foreign' genes would be likewise affected. Therefore, we re-analyzed our previously published small RNA sequencing data, obtained from *set-25* mutants (*Lev et al., 2017*). However, among the targets of these differentially expressed endo-siRNAs, we could not detect striking changes (fold change >1.2) in endo-siRNAs that target transposons and repetitive elements in *set-25* mutants (*Figure 3A*, left panel). In contrast, a subset of endo-siRNAs that target 279 different protein-coding genes was found to exhibit significant changes in *set-25* mutants (adj.p <0.1, DESeq2 *Figure 3A*, right panel). To understand why these small RNAs are uniquely affected by SET-25, we characterized this group and the endo-siRNAs that target them. To compare the endo-siRNA pools that depend on these two H3K9 tri-methyltrasferases, we re-analyzed the recently published small RNA-seq data obtained from *set-32* mutants (*Kalinava et al., 2018*).

Since in *set-25* the loss of exogenous siRNAs coincided with the loss of heritable RNAi-induced H3K9me3 (*Lev et al., 2017*), we first tested whether genes that were differentially targeted by endo-siRNAs in *set-25* mutants were also marked by H3K9me3. By examining publicly available H3K9me3 data (*McMurchy et al., 2017*), we found that the 151 genes that lost the endo-siRNAs that target them in *set-25* mutants were robustly marked by H3K9me3 in wild type animals (*Figure 3B*). We also found that in contrast, the 128 genes that had *increased* endo-siRNA levels that target them in *set-25* and mutants were not significantly marked by H3K9me3 (*Figure 3—figure supplement 1A*). By analyzing an available mRNA-seq dataset (*Klosin et al., 2017*), we also found a significant enrichment for genes that were *upregulated* (at the mRNA level) in *set-25* mutants amongst the list of SET-25-dependent endo-siRNA targets (1.93-fold enrichment, 18/151 genes, p-value=0.006). This suggests that endo-siRNAs that depend on SET-25 silence targeted gene. Recently, Kalinava et al. sequenced endo-siRNAs from *set-32* mutants (*Kalinava et al., 2018*). Our analysis show that the 337 genes that had reduced levels of endo-siRNAs (fold change >2) in *set-32* mutants, were also significantly marked by H3K9me3 (*Figure 3B*). As expected, these genes showed lower levels of H3K9me3 in *set-32* mutants (*Figure 3—figure supplement 1B*), and genes having *increased* levels of endo-siRNAs were not significantly marked by H3K9me3 (*Figure 3—figure supplement 1A*). Together, these results support the hypothesis that H3K9me3 methyltransferases directly support the biogenesis of silencing endo-siRNAs by tri-methylating the H3K9 histones of the endo-siRNAs targeted genes.

Next we examined whether genes that display altered endo-siRNAs levels in *set-25* and *set-32* mutants are expressed in specific tissues. Genes that had significantly reduced levels of endo-siRNAs targeting them in *set-25* or in *set-32* mutants exhibited significant, but modest, enrichment for expression in the germline (*Figure 3C* and *Figure 3—figure supplement 1C*). No significant *enrichment* was found for other tissues (*Figure 3C* and *Figure 3—figure supplement 1C*).

To identify the small RNA pathways which are affected by *set-25* and *set-32*, we tested whether the differentially expressed endo-siRNAs depend on particular argonautes, or associate with specific biosynthesis or functional pathways (*Figure 3D*). It was previously suggested that the CSR-1 argonaute carries heritable endo-siRNAs that mark endogenous genes (*Claycomb et al., 2009*), while the HRDE-1 argonaute carries heritable endo-siRNAs that silence foreign or aberrant elements, whose expression could be deleterious, such as transposons (*Luteijn et al., 2012*; *Rechavi, 2014*; *Shirayama et al., 2012*). A strong and significant enrichment (*Figure 3D* and *Figure 3—figure supplement 1C*) was found for endo-siRNAs which are carried in the germline by the argonautes WAGO-1 (*Gu et al., 2009*) and HRDE-1, which is required for inheritance of exogenous siRNAs (*Buckley et al., 2012*). Both argonautes were found to be involved in gene silencing (*Buckley et al., 2012*; *Gu et al., 2009*). Nevertheless, some of the targets of HRDE-1-bound endo-siRNAs are expressed in the germline (*Figure 3—figure supplement 2A*). This may explain the concurrent

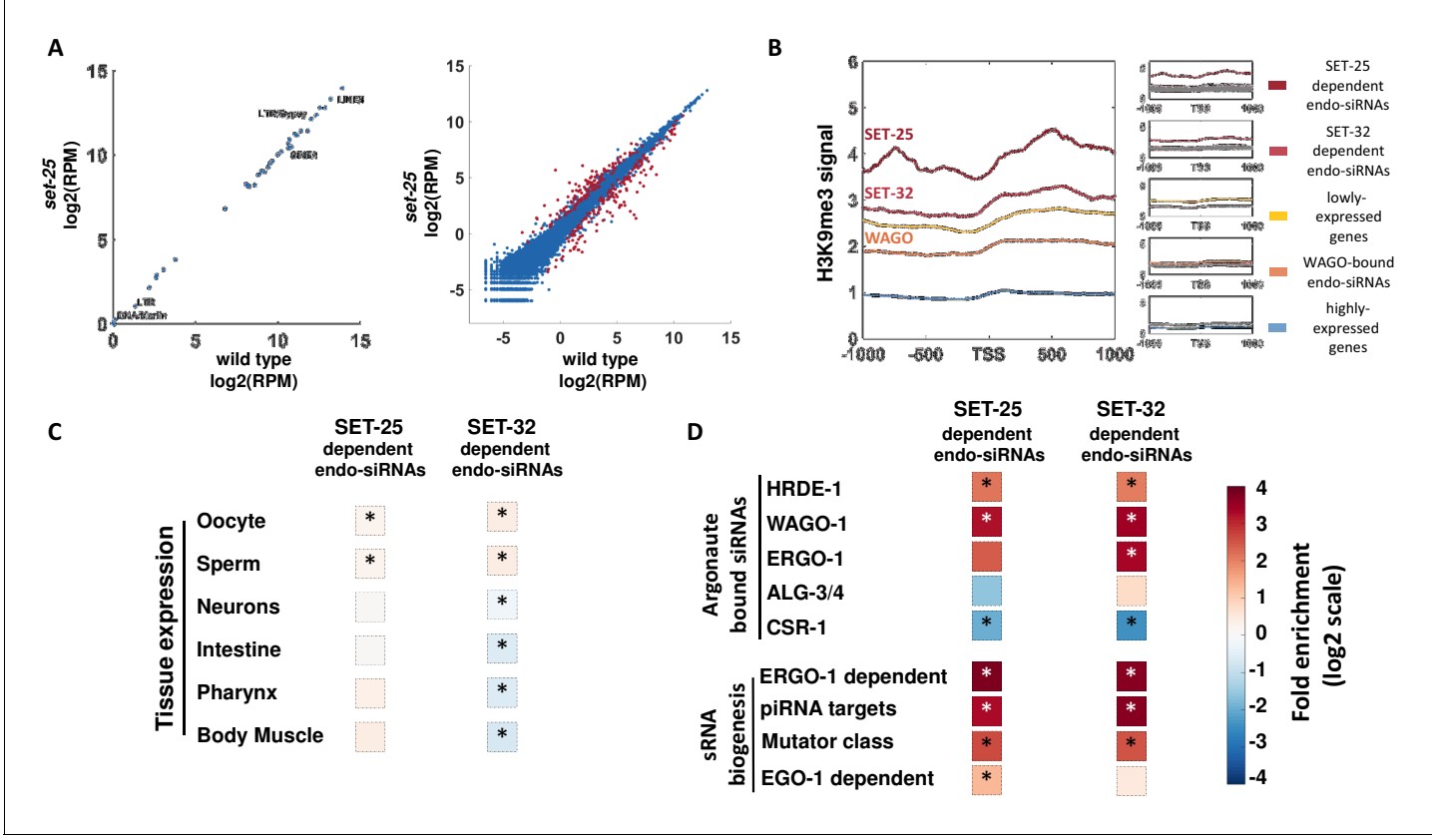

**Figure 3.** A genome-wide analyses of endo-siRNAs that depend on SET-25 or SET-32. (A) An expression analysis of endo-siRNAs targeting transposons and repetitive elements classes (left panel) or protein-coding genes (right panel). Shown are the expression values as log2 of number of Reads per Million (RPM) in *set-25* mutants (y-axis) compared to wild type animals (x-axis). Gene targets of endo-siRNAs, which display significant differential expression (analyzed with Deseq2, adjusted p-value<0.1) are marked in Red (B) An analysis of H3K9me3 signals (based on published data from *McMurchy et al., 2017*) on differet sets of gens: highly expressed genes (top 10%, Blue), lowly expressed genes (top 10%, Yellow) and gene targets of endo-siRNAs that depend on SET-25 (based on *Lev et al., 2017*, Red), SET-32 (based on *Kalinava et al., 2018*, Dark Red) or endo-siRNAs associated with WAGOs (HRDE-1,WAGO-1, and ERGO-1, Light Red). H3K9me3 signal is aligned according to gene's Transcription Start Sites (TSS), and the regions of 1000 base pairs upstream and downstream of the TSS are shown on the x axis. The y axis shows the averaged signal of the H3K9me3 modification as a function of distance from the TSS. For statistical analysis, control data sets (shown in Gray) were created by sampling the H3K9me3 levels of randomly selected gene sets of the same size as the examined gene list. (C and D) An enrichment analysis of genes with significantly lowered levels of endo-siRNAs targeting them in *set-25* and *set-32* mutants compared to wild type. Fold enrichment values (log2 scale) are color coded. (C) An enrichment analysis for expression in specific tissues. (D) An enrichment analysis for different small RNA pathways. The p-values were calculated using 10,000 random gene sets identical in their size to the examined endo-siRNA-target gene list. Asterisk denotes statistically significant enrichment values (p-value<0.05).

DOI: https://doi.org/10.7554/eLife.40448.007

The following figure supplements are available for figure 3:

**Figure supplement 1.** An analysis of genes targeted by endo-siRNAs upregulated in *set-25* and *set-32* mutants.

DOI: https://doi.org/10.7554/eLife.40448.008

**Figure supplement 2.** An analysis of overlap between different endo-siRNA target gene sets.

DOI: https://doi.org/10.7554/eLife.40448.009

enrichment for both germline-expressed genes and targets of HRDE-1-bound endo-siRNAs amongst the gene targets of endo-siRNAs that depend on SET-25 or SET-32. A significant enrichment was also found for Mutator pathway small RNAs (*Zhang et al., 2011*), ERGO-1-dependent small RNAs, and putative piRNA targeted genes (*Bagijn et al., 2012*). On the contrary, a significant *depletion* was found for genes known to be targeted by CSR-1-carried small RNAs, a pathway that was suggested to support the expression of targeted genes (*Claycomb et al., 2009*; *Shen et al., 2018*). The helicase EMB-4 (*Akay et al., 2017*; *Tyc et al., 2017*) was shown to preferably bind introns of genes targeted by CSR-1; We could not detect a significant enrichment for genes whose introns are bound

by EMB-4 (fold change = 1.07 and 0.79, p-value=0.26 and 0.002, for endo-siRNAs dependent on SET-25 or SET-32, respectively). All together, these results suggest that H3K9 methyltransferases are required for the maintenance of a specific sub-class of HRDE-1 and WAGO-1 small RNAs, that are associated with the Mutator and piRNA pathways, and that target protein-coding genes (*Figure 3—figure supplement 2B*).

## Endo-siRNAs that depend on H3K9me3 methyltransferases target a distinctive subset of newly evolved genes

What distinguishes the target genes of endo-siRNAs that depend on SET-25 and SET-32 methyltransferases? It was recently found that periodic A/T (PATC) sequences can shield germline genes from piRNA-induced silencing and allow germline expression of genes in H3K9me3-rich genomic regions (*Frøkjær-Jensen et al., 2016*; *Zhang et al., 2018*). Fittingly, we found that genes targeted by SET-25-dependent and SET-32-dependent endo-siRNAs exhibit a moderate (~9–13% in median values) but significant reduction in PATC density compared to all protein coding genes (*Figure 4A*, p-value = 0.0026 and 0.0011 for SET-25 and SET-32, respectively). This feature is not general for genes targeted by WAGOs (Worm-specific Argonautes, HRDE-1, WAGO-1 and ERGO-1) associated endo-siRNAs, since these targeted genes have a higher PATC density (*Figure 4A*, 10% increase in average values, p-value=0.034). In addition, genes targeted by endo-siRNAs that are *increased* in *set-25* or *set-32* mutants exhibit significantly increased PATC density (*Figure 4—figure supplement 1A*). However, we posit that this feature is unlikely to be sufficient for distinguishing between *oma-1* and *gfp*, since the *oma-1* gene has a very low PATC density (*Figure 4—figure supplement 1B*).

The lists of genes which are targeted by SET-25- and SET-32-dependent endo-siRNAs were enriched for genes targeted by ERGO-1-dependent endo-siRNAs (*Figure 3D*). Many of the genes that are targeted by ERGO-1-bound endo-siRNAs are duplicated genes (*Fischer et al., 2011*; *Vasale et al., 2010*). Accordingly, we found an enrichment for duplicated genes amongst the genes that had reduced endo-siRNA levels targeting them in *set-25* and *set-32* mutants (*Figure 4B*). An additional characteristic of the set of genes targeted by ERGO-1 endo-siRNAs is an enrichment for poorly conserved genes, that have fewer introns, and possess splicing site sequences that diverge from the consensus sequence (*Fischer et al., 2011*; *Newman et al., 2018*). It was recently suggested that these poorly conserved genes are targeted for silencing because their aberrant or"non-self-like' splicing signals are detected by the splicing machinery (*Newman et al., 2018*).

Therefore, we examined whether the targets of the endo-siRNAs that depend on SET-25 or SET-32 can be distinguished by their splicing signals. The changes in the endo-siRNA pool in mutants of small nuclear ribonucleoprotein-associated protein RNP-2/U1A (*rnp-2*) mirrored the endo-siRNA changes found in *set-25* mutants (*Figure 4C*), but not that of *set-32* mutants (*Figure 4—figure supplement 2A*). We also found that genes targeted by SET-25-dependent- but not SET-32-dependent endo-siRNAs bear fewer introns (*Figure 4D*, median of 3 and 4 compared to 4 of all protein coding genes, p-values=0.0047, and 0.42 for SET-25 and SET-32, respectively). No significant differences in the length of the coding sequences were found, hence, the difference in intron number does not simply derive from differences in gene lengths (*Figure 4—figure supplement 2B*, p-value = 0.8673). The lists of genes targeted by SET-25-dependent or SET-32-dependent endo-siRNAs were enriched with genes shown to be targeted by intron-targeting small RNAs (*Figure 4—figure supplement 2C and D*). We could not find, however, small RNAs aligning to the introns of the *gfp* transgene that we studied (*Figure 4—figure supplement 2E*, in most cases endo-siRNAs target only exons). We also did not find significant differences in the splicing motif divergence score (obtained from *Newman et al., 2018*). Since splicing also directly affects the RNAi machinery untangling its role in endogenous RNAi is challenging (*Newman et al., 2018*). In summary, splicing may be one of the factors that contribute to distinguishing genes targeted by SET-25-dependent endo-siRNAs, but not by SET-32-dependent endo-siRNAs.

In contrast, in the sets of genes targeted by either SET-25-dependent or SET-32-dependent small RNA we found a significant enrichment for *newly evolved genes* (*Figure 4B*, fold-change = 2.57 and p-value<0.0001 for both SET-25- and SET-32-dependent endo-siRNAs targets, respectively). We define *newly evolved genes* here as genes which had no orthologs outside *C. elegans* (35/151 and 78/337 of genes targeted by SET-25- or SET-32-dependent endo-siRNAs, respectively). Concordantly, in the same gene sets we also found a significant *depletion* for nematode-conserved genes

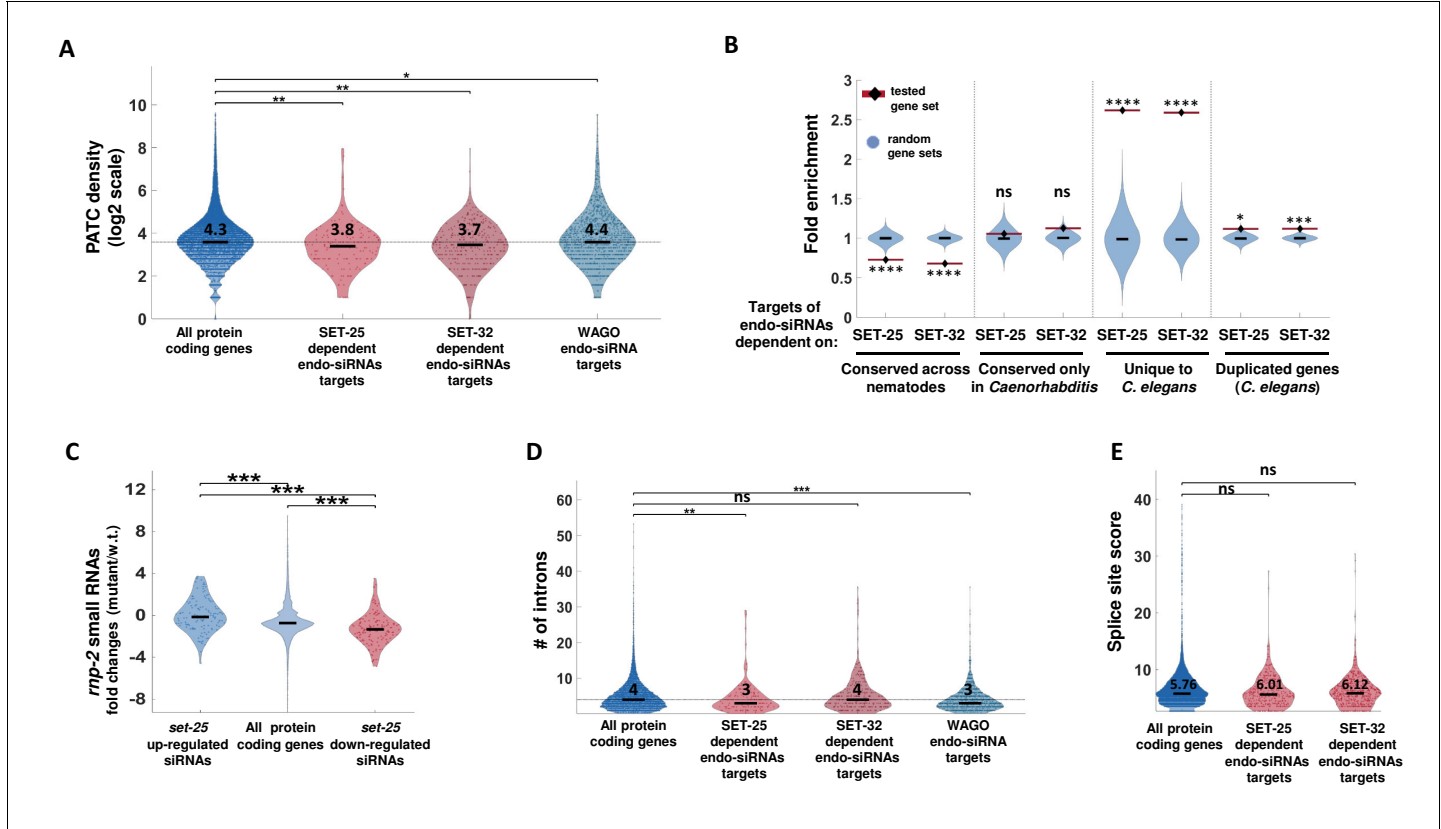

**Figure 4.** SET-25-dependent endo-siRNAs target newly evolved genes. (**A**) A PATC density analysis for SET-25 and SET-32-dependent endo-siRNAs gene targets. The PATC density values (obtained from *Frøkjær-Jensen et al., 2016*) are presented for all protein-coding genes, gene targets of endo-siRNAs that depend on SET-25 or SET-32 and gene targets of endo-siRNAs associated with WAGO small RNA pathways (HRDE-1,WAGO-1 or ERGO-1). **p-value<0.005, *p-value<0.05, Wilcoxon rank sum test. For clarity of display, values are shown in log2 scale (after addition of 1). The median (black line) and average levels (numbers) of PATC density levels of each plot are indicated (log2 scale). (**B**) An enrichment analysis of genes conserved at different levels and duplicated genes amongst gene targets of SET-25- and SET-32-dependent endo-siRNAs. The gene sets were generated based on the homology field in WormBase that details the orthologs and paralogs of each nematode gene. We defined a duplicated gene as a gene that has a paralog in *C. elegans*. We define genes unique to *C. elegans* as genes that lack an ortholog amongst the nematode species we examined (see Materials and methods). For statistical analysis, control enrichment values were obtained from 10,000 random gene sets with the same size as the examined endo-siRNA-target gene list. ****p-value<0.0001,***p-value<0.001, *p-value<0.05 (**C**) An analysis of endo-siRNAs fold changes in *rnp-2* mutants for genes targets of endo-siRNAs downregulated or upregulated in *set-*25 mutants or all genes. All p-values<0.001, Wilcoxon rank sum test. (**D**) An analysis of intron numbers of gene targets of SET-25- and SET-32-dependent endo-siRNAs and WAGO-associated endo-siRNAs compared to all protein-coding genes. In cases of genes that have more than one transcript, the average intron value is used. The median intron number of each plot is indicated (log2 scale). ***p-value<0.001,**p-value<0.005, Wilcoxon rank sum test. (**E**) An analysis of splicing motif divergence score (based on *Newman et al., 2018*) of gene targets of SET-25 and SET-32-dependent endo-siRNAs, WAGO associated endo-siRNAs and all protein-coding genes. The median score levels of each plot are indicated. p-value>0.05, Wilcoxon rank sum test.

DOI: https://doi.org/10.7554/eLife.40448.010

The following figure supplements are available for figure 4:

**Figure supplement 1.** An analysis PATC density of the *oma-1* gene and of targets of endo-siRNAs upregulated in *set-25* and *set-32* mutants.
DOI: https://doi.org/10.7554/eLife.40448.011

**Figure supplement 2.** Gene targets of endo-siRNAs that depend on SET-25 or SET-32 are preferably targeted by small RNAs on intron-exon junctions compared to all protein coding genes.
DOI: https://doi.org/10.7554/eLife.40448.012

**Figure supplement 3.** An analysis of endo-siRNAs targeting and H3K9me3 levels of newly evolved genes.
DOI: https://doi.org/10.7554/eLife.40448.013

(*Figure 4B*). Importantly, the sets of genes targeted by SET-25-dependent endo-siRNAs and SET-32-dependent endo-siRNAs show very small overlap (25 out of 465 genes). Thus, while SET-25 and SET-32 are required for the maintenance of endo-siRNAs that target different genes, the characteristics of these genes are very similar, that is they are distinctively newly evolved genes that have slightly lower levels of PATC sequences. Although the changes in PATC density and intron numbers that distinguish these target genes are moderate, it is possible that the cumulative effect of these small differences may result in the exposure of foreign genes that need to be silenced.

In general, we find that certain sub-classes of endo-siRNA, such as ERGO-1 and HRDE-1 bound small RNAs, target gene sets enriched for newly evolved genes (*Figure 4—figure supplement 3*). The significant enrichment for newly evolved genes among SET-25- and SET-32-dependent endo-siRNAs is maintained, however, even after excluding genes that are also targeted by HRDE-1, ERGO-1, WAGO-1 or Mutator endo-siRNAs (SET-25: 59/151 genes are not shared, fold enrichment = 2.97, p-value=0.0001, SET-32: 153/337 genes are not shared, fold-enrichment = 1.89, p-value=0.0012). Thus, the enrichment of newly evolved genes amongst the targets of SET-25- and SET-32-dependent endo-siRNAs is not simply due to a general preference for newly evolved genes by endo-siRNA pathways. Further, we find that newly evolved genes are marked by higher levels of H3K9me3 in comparison to the average level of H3K9me3 on protein coding genes (*Figure 4—figure supplement 3*). Likewise, in the absence of RNAi, in wild-type animals, *gfp*, the example for a foreign (non-nematode) gene that we investigated, has higher levels of H3K9me3, in comparison to the well-conserved *oma-1* gene (*Figure 2B*). The fact that across the genome SET-25-dependent- and SET-32-dependent endo-siRNAs target newly evolved and H3K9me3 methylated genes (*Figure 3B* and *Figure 4B*), may explain why inheritance of RNAi responses raised against *gfp*, but not *oma-1*, depends on SET-25 and SET-32 (*Figure 1*).

In summary, our experiments reveal a specific role for histone modifications in small RNA inheritance. While in S. *pombe* and A. *thaliana* a feedback between H3K9me3 and small RNAs was suggested to be required for silencing, the worm's RNAi inheritance machinery may use H3K9me3 as a mark that distinguishes genes identified as 'new'. Since newly evolved genes can be disruptive, small RNAs survey these H3K9me3-flagged elements transgenerationally.

## Discussion

Our study began from an investigation of a perplexing asymmetry in the requirement of specific H3K9 methyltransferases for heritable silencing of the endogenous gene *oma-1* and the 'foreign' gene *gfp*. Single mutants of *set-25* and *set-32* and the *met-2;set-25;set-32* triple mutant displayed different heritable dynamics when either the *gfp* or the *oma-1* gene were targeted by RNAi. These results are not unique to the specific *gfp* transgene that was tested, since similar observations have been made with other transgenes (*Klosin et al., 2017*; *Lev et al., 2017*; *Shirayama et al., 2012*; *Spracklin et al., 2017*).

Unlike mutations in these histone methyltransferases, which negatively affect heritable silencing of *gfp*, but not *oma-1*, mutations in genes required for small RNA inheritance negatively affect heritable silencing of both *oma-1* and *gfp*. For example, the argonaute HRDE-1 is required for inheritance of RNAi responses against both genes (*Ashe et al., 2012*; *Buckley et al., 2012*; *Kalinava et al., 2017*; *Shirayama et al., 2012*; *Weiser et al., 2017*). The fact that heritable RNAi responses aimed at different genes are affected by different proteins should be taken into account when studying transgenerational inheritance. Specifically, when screening for genes that affect such inheritance, one must acknowledge that heritable silencing of different targets requires different chromatin modifiers.

Future studies will hopefully reveal why some recently evolved genes, but not others, display high levels of H3K9me3 (in the absence of RNAi), and are targeted by endo-siRNAs. Recent studies examined why transgenes are sensitive to silencing by synthetic piRNAs, while endogenous germline expressed genes, including *oma-1*, are not. This protection was suggested to be conferred at least in part by PATC sequences, and to be independent of the genomic location of the gene (*Zhang et al., 2018*). PATC sequences were previously shown to allow expression of transgenes in the germline in heterochromatic areas (*Frøkjær-Jensen et al., 2016*). Similarly, our analysis revealed that the gene targets of SET-25-dependent and SET-32-dependent endo-siRNAs have lower levels of PATC density (*Figure 4A*). However, the *oma-1* gene does not possess many PATC sequences

(*Figure 4—figure supplement 1B*). An additional theory suggested that an intrinsic unknown coding-sequence feature confers resistance to silencing by piRNAs. Seth et al. have studied why a fusion between *oma-1* and *gfp* can trans-activate silenced *gfp* transgenes (an effect known as 'RNAa', (*Seth et al., 2013*)). While unique 'protective' sequence features were not described in that work, the authors showed that an unknown coding-sequence feature, not related to the codon usage or the translation of the protein, grants the *oma-1* gene with its ability to activate silenced transgenes (*Seth et al., 2018*). It is possible that the gene targets of SET-25- dependent and SET-32-dependent small RNAs that we describe here have unique intrinsic sequences that distinguish them as well. The different requirement of methyltransferases for heritable silencing of some genes but not others may be related to such intrinsic sequence features. Alternatively, it is possible, as was suggested in the past, that new genes are silenced because they are not licensed transgenerationally by heritable small RNAs for expression (*Claycomb et al., 2009*; *Shen et al., 2018*). If this is the case, future studies will hopefully reveal how such license is granted (See *Figure 5* for Scheme).

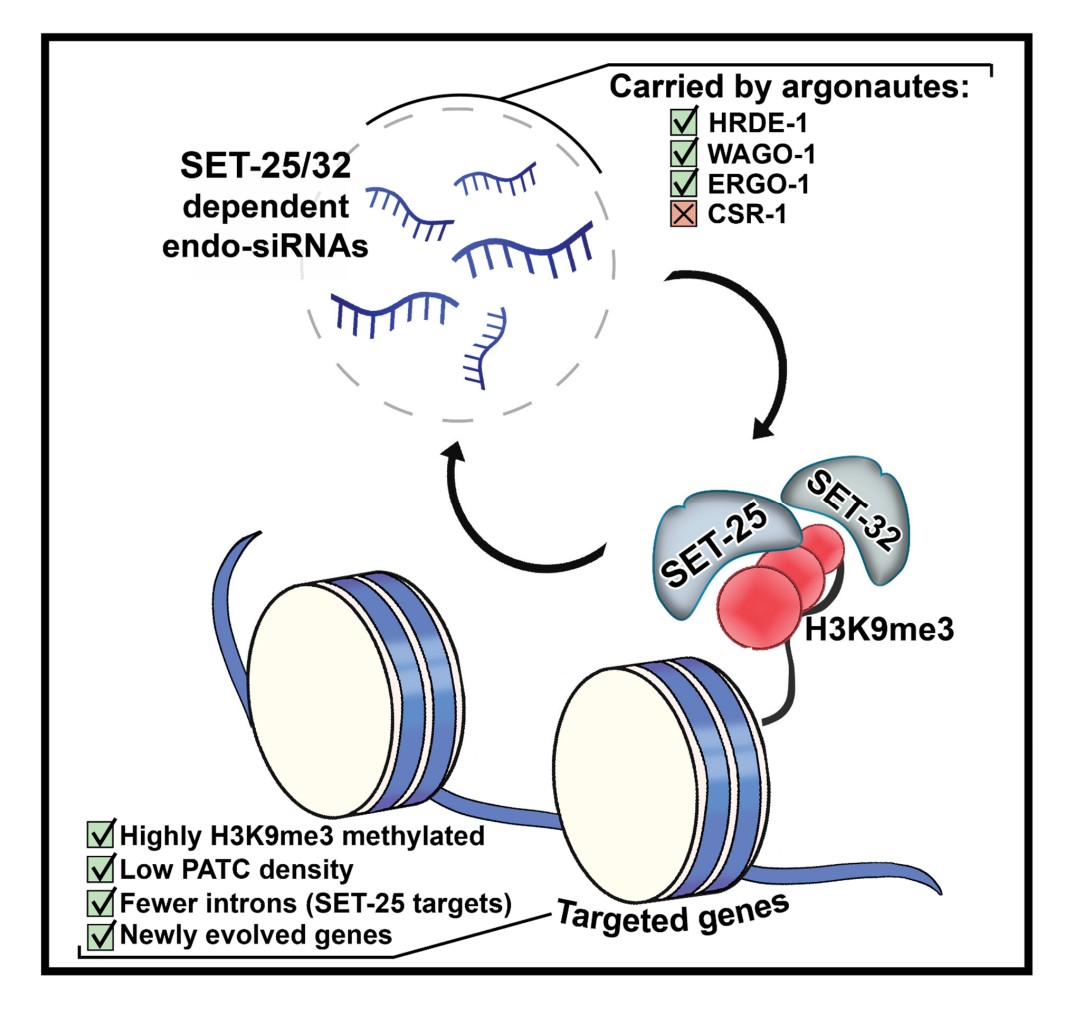

**Figure 5.** Scheme characterizing H3K9me3 methyltransferase-dependent endo-siRNAs and their targets. SET-25-dependent and SET-32-dependent endo-siRNAs are enriched with small RNAs known to be carried by the argonautes HRDE-1, WAGO-1 and ERGO-1 but not CSR-1. SET-25-dependent and SET-32-dependent endo-siRNAs targets are enriched with newly evolved genes, have fewer PATC sequences, and are marked with higher levels of H3K9me3. Targets of SET-25-dependent but not SET-32-dependent endo-siRNAs bear fewer introns.
DOI: https://doi.org/10.7554/eLife.40448.014

# Materials and methods

**Key resources table**

| Reagent type | Designation | Source of reference | Identifiers | Additional information |
|---|---|---|---|---|
| Strain (*E. coli*) | OP50 | (*Brenner, 1974*) | Op50 RRID:WB-STRAIN:OP50 | |
| Strain (*E. coli*) | anti-*gfp* RNAI bacteria | This study | | |
| Strain (*E. coli*) | anti-*oma-1* RNAi bacteria | Ahringer RNAi library (*Kamath and Ahringer, 2003*) | 4-4-3-C01 | |
| Strain (*E. coli*) | anti-*sup-35* RNAi bacteria | Vidal RNAi library (*Rual et al., 2004*) | 11006-E2 | |
| Strain (*E. coli*) | anti-*fog-2* RNAi bacteria | Vidal RNAi library (*Rual et al., 2004*) | 10011-C3 | |
| Strain (*C. elegans*) | N2 | CGC | N2 RRID:WB-STRAIN:N2_ (ancestral) | |
| Strain (*C. elegans*) | *set-32(ok1457)* ; *oma-1(zu405)* | CGC | BFF25 | |
| Strain (*C. elegans*) | *mjIs134[pmex-5::gfp::h2b::tbb-2]* | Erik Miska's lab (Univeristy of Cambridge) | SX1263 | |
| Strain (*C. elegans*) | *oma-1(zu405)* | CGC | TX20 | |
| Strain (*C. elegans*) | *pha-1(e2123)* | CGC | GE24 | |
| Strain (*C. elegans*) | *set-32(ok1457); mjIs134[pmex-5::gfp::h2b::tbb-2]* | This study | BFF24 | |
| Strain (*C. elegans*) | *met-2(n4256); set-25(n5021); set-32(ok1457); mjIs134[pmex-5::gfp::h2b::tbb-2]* | This study | BFF26 | |
| Strain (*C. elegans*) | *met-2(n4256); set-25(n5021); set-32(ok1457); oma-1(zu405)* | This study | BFF27 | |
| Strain (*C. elegans*) | *met-2(n4256); set-32(ok1457); mjIs134[pmex-5::gfp::h2b::tbb-2]* | This study | BFF28 | |
| Antibody | H3K9me3 | Abcam | RRID:AB_306848 | |
| Software, algorithm | GraphPad Prism | https://www.graphpad.com/scientific-software/prism/ | RRID:SCR_002798 | |
| Software, Algorithm | Image J | Opensource: https://imagej.nih.gov/ij/ | RRID:SCR_003070 | |
| Software, Algorithm | MATLAB MathWorks | https://www.mathworks.com/ | RRID:SCR_016651 | |

## Cultivation of the worms

Standard culture techniques were used to maintain the nematodes on nematode growth medium (NGM) plates seeded with OP50 bacteria. Extreme care was taken to avoid contamination or starvation, and contaminated plates were discarded from the analysis.

## RNAi bacteria

HT115 *Escherichia coli* strains expressing dsRNAs were used: anti-*oma*-1 RNAi bacteria were obtained from the Ahringer RNAi library (*Kamath and Ahringer, 2003*). Anti-*fog-2* and anti-*sup-35* RNAi were obtained from the Vidal RNAi library (*Rual et al., 2004*). For the sequence of the anti-*gfp* RNAi see supplemental data.

## RNAi experiments

RNAi HT115 *E.coli* bacteria were incubated in lysogeny broth (LB) containing Carbenicillin (25 µg/mL) at 37°C overnight with shaking. Bacterial cultures were seeded onto NGM plates containing iso-propyl β-D-1-thiogalactopyranoside (IPTG; 1 mM) and Carbenicillin (25 µg/mL) and grown overnight in the dark at room temperature. Five L4 animals were placed on RNAi bacteria plates and control empty-vector bearing HT115 bacteria plates and maintained at 20°C for 2 days and then removed. The progeny hatching on these plates was termed the P0 generation. In the next generations the worms were grown on *E.coli* OP50 bacteria. For anti-*gfp* RNAi experiments, four L4 animals were placed on plates for two days to lay the next generation. In every generation approximately 60 one day adult worms were collected and photographed per condition (see below). For anti-*oma-1* experiments, in each generation twelve individual L4 staged worms were placed in individual wells of a twelve well plate. Four days later the number of fertile worms was assessed (at least one progeny) and 12 individual L4 progeny worms were chosen from the most fertile well to continue to the next generation. For and anti-*sup-35* RNAi experiments, in each generation 12 individual L4 staged worms were placed in individual wells of a 12-well plate. Two days later the adult worms were removed. Two days later the number of developing worms was counted and twelve individual L4 progeny worms were chosen from the well with the highest number of developing progeny to continue to the next generation. For and anti-*fog-2* RNAi experiments, five L4 worms were crossed on RNAi bacteria for 24 hr. The crossed worms were transferred to fresh RNAi bacteria plates. In each generation, five resulting L4 progeny from the cross were crossed on control bacteria plates and ~40 L4 worms were picked to control bacteria plates and photographed a day later. The number of sterile worms with stacked oocytes was assayed.

## Germline GFP expression analysis

Percentage silencing analysis: for each condition, around 60 animals were mounted on 2% agarose slides and paralyzed in a drop of M9 with 0.01% levamisole/0.1% tricaine. The worms were photographed with 10x objective using a BX63 Olympus microscope (Exposure time of 200 ms, and gain of 2). The images were analyzed with ImageJ2 software, and the percentage of worms lacking any observable germline GFP signal was calculated.

GFP expression level analysis: for each condition, the GFP fluorescence level of the background and of oocyte nuclei of at least 30 worms was calculated using ImageJ2.

CTCF value was calculated as follows: CTCF = Integrated density of selected object X – (area of selected object X * mean fluorescence of background readings). The obtained CTCF value was normalized to the average CTCF value obtained from photographs of control animals of the same genotype, generation and age which were fed on control plates.

## Chromatin immunoprecipitation

Chromatin immunoprecipitation experiments were conducted as described in *Lev et al. (2017)*. For anti-H3K9me3 ChIP experiments the abcam, ab8898 antibodies were used.

## qPCR reactions

All Real time PCR reactions were performed using the KAPA SYBR Fast qPCR and run in the Applied Biosystems 7300 Real Time PCR System.

The primer sequences used in qRT-PCR:
*gfp* set #1 FOR: ACACAACATTGAAGATGGAAGC
*gfp* set #1 REV: GACAGGTAATGGTTGTCTGG
*gfp* set #2 FOR: GTGAGAGTAGTGACAAGTGTTG
*gfp* set #2 REV: CTGGAAAACTACCTGTTCCATG
*oma-1* set#1 FOR: AACTTTGCCCGTTTCACC

*oma-1* set#1 REV: TCAAGTTAGCAGTTTGAGTAACC
*oma-1* set#2 FOR: TTGTTAAGCATTCCCTGCAC
*oma-1* set#2 REV: TCGATCTTCTCGTTGTTTTCA
(The above primer set was adapted from *Spracklin et al., 2017*)
*dpy-28* FOR: CTGATGGATCCAGAGTTGG
*dpy-28* REV: CTGCTATACGCATCCTGTTC
*eft-3* FOR: CCAACATGATTAGTCAGATGACC
*eft-3* REV: CTAGGAGTTAGATGTGCAGG.

## Information on the sequencing libraries analyzed in this paper

All the studied publicly available sequencing libraries were prepared from synchronized young adult worms grown at 20 degrees (*Kalinava et al., 2018*; *Klosin et al., 2017*; *Lev et al., 2017*; *McMurchy et al., 2017*). For more information see the original publications and GEO information: A. *set-25* small RNAs (*Rechavi and Lev, 2017*; GEO accession: GSE94798), B. *set-25* mRNA (*Klosin et al., 2017*; GEO accession: GSE83528), C. *set-32* small RNAs (*Kalinava et al., 2018*; GEO accession: GSE117662, the *set-32* (red11) allele data were used)., D. *set-32* and wild type H3K9me3 ChIP-seq (*Kalinava et al., 2018*; GEO accession: GSE117662). E. wild type H3K9me3 ChIP-seq (*McMurchy et al., 2017*; GEO accession: GSE87524).

## Bioinformatic genome-wide endo-siRNAs analysis

Small RNA analysis was conducted as previously described (*Lev et al., 2017*). Briefly, adapters were cut from the reads using Cutadapt (*Martin, 2011*). Reads that were not cut or were less than 19 bp long, were removed. The quality of the libraries was assessed by FastQC (http://www.bioinformatics.babraham.ac.uk/projects/fastqc/). Reads were mapped to the *C. elegans* genome (WS235) using Bowtie2 (*Langmead and Salzberg, 2012*). In total 31,053,062, 21,913,420 and 18,372,739 reads were mapped in the three wild type biological repeats and 21,258,241, 19,925,004 and 21,391,091 reads were mapped in the three *set-25* biological repeats. The mapped reads were then counted using the python script HTseq_count (*Langmead and Salzberg, 2012*) using. gff feature file from wormbase.org (version WBcel235). Differential expression was analyzed using DESeq2 (*Love et al., 2014*). p-adjusted value <0.1 was regarded as statistically significant. The *set-32* data from Kalinava et al (*Kalinava et al., 2018*) GEO was analyzed in a similar fashion. The reads the 5' barcode and 3' linker were trimmed using Cutadapt (*Martin, 2011*), in accordance the information supplied by Kalinava et al in GEO (accession number: GSE117662). Next, reads were filtered to lengths of 20–23 bp and aligned (not allowing mismatches) to the *C. elegans* genome (ce10) by Shortstack (*Axtell, 2013*). In total 1,342,884 and 1,006,568 reads were mapped in the wild type and *set-32(red11)* small RNA samples, respectively. The reads mapping to each genomic were counted by HTseq_count (*Langmead and Salzberg, 2012*). Since one biological sample was available, significantly altered small RNAs were defined as genes having fold change of larger than 2 (up-regulated) or smaller than 0.5 (downregulated).

## Bioinformatic genome-wide analysis of H3K9me3 signal

For analysis of H3K9me3 signal on different genes in wild type worms, the processed H3K9me3 data (aligned and normalized) from the McMurchy et al. study was used (*McMurchy et al., 2017*; GEO accession GSE87524). The shown H3K9me3 signal represents the averaged H3K9me3 signal in two replicates of young adults. For analysis of the H3K9me3 levels in wild type and *set-32* mutants the raw data from the Kalinava et al. study was used (*Kalinava et al., 2018*; GEO accession: GSE117662). The raw data were analyzed in a similar fashion to the analysis conducted by McMurchy et al. Briefly, adaptors were trimmed using Cutadapt (*Martin, 2011*) and aligned using Bowtie2 (*Langmead and Salzberg, 2012*). H3K9me3-enriched regions were identified using MACS2 (*Lupien et al., 2008*) and the H3K9me3 signal was corrected for biases using BEADS (*Cheung et al., 2011*).

## Bioinformatic mRNA expression analysis

Processed files with raw counts of reads mapping to each gene were downloaded from GEO (*Klosin et al., 2017*; GEO accession: GSE83528). Differential expressed genes were detected using DESeq2 (adjusted p-value<0.1).

## Bioinformatic gene enrichment analysis

The enrichment values denote the ratio between (A) the observed representations of a specific gene set within a defined differentially expressed genes group, to (B) the expected one, that is the representation of the examined gene set among all protein-coding genes in *C. elegans*. The analysis was done for 15 gene sets: (1) 7727 genes enriched in oocytes gonads (*Ortiz et al., 2014*) and 9012 genes enriched in spermatogenic gonads (*Ortiz et al., 2014*); we excluded genes with expression lower than 1 RPKM(2) 11427 genes expressed in isolated neurons (*Kaletsky et al., 2016*). (3) 7176 genes expressed in intestine (*Gerstein et al., 2010*) (4) 2957 genes expressed in pharynx (*Gerstein et al., 2010*) (5) 2526 genes expressed in body muscle (*Gerstein et al., 2010*) (6) 4146 targets of CSR-1 (*Claycomb et al., 2009*) (7) 1478 targets of HRDE-1 (*Buckley et al., 2012*) (8) 87 targets of WAGO-1 (*Gu et al., 2009*) (9) 399 targets of ALG-3/4 class small RNAs (*Conine et al., 2010*) (10) 1823 targets of mutator class small RNAs (11) 721 EGO-1 dependent small RNA gene targets (*Maniar and Fire, 2011*), (12) 23 gene targets of small RNAs up-regulated in *ego-1* mutants (*Maniar and Fire, 2011*), (13) 49 genes targeted by 26G-RNAs enriched in ERGO-IP (*Vasale et al., 2010*) (14) 77 genes depleted of 22G-RNAs in *ergo-1* mutants (*Vasale et al., 2010*), and (15) 348 putative piRNA gene targets (*Bagijn et al., 2012*). The putative piRNA gene targets were defined as genes for which, in at least one transcript, the ratio of the # 22G-RNA reads at piRNA target sites between wild type to *prg-1* is at least 2 (linear scale). Note that the indicated number above achieved after intersection between the various published data sources and the records appears in the *.gff file used by us.

The enrichment value of a given gene set i in differentially expressed gene targeting small RNAs was calculated using the following formula:

$$Enrichment = \frac{Observed}{Expected} = \left( \frac{fraction\ of\ genes\ belong\ to\ the\ i-th\ set\ among\ differentially\ expressed\ STGs}{fraction\ of\ genes\ belong\ to\ the\ i-th\ set\ among\ all\ the\ genes} \right)$$

Obtaining the observed-to-expected ratios, we then calculated the corresponding p-values using 10,000 random gene groups identical in size to that of the examined group of differentially expressed genes. Next, the enrichment values of the random sets are ranked and the p-value is determined by the ranking of the examined gene set amongst the ranking of all enrichment values of the random sets.

## Gene sets by conservation

The classification of gene sets by conservation was done by mining the 'Homology' field of all the *C. elegans* protein-coding genes in WormBase (www.wormbase.com). We defined the following three gene sets (*Figure 4B*):

1. Unique to *C. elegans* – *C. elegans* genes which have no orthologues gene in any of the following species: *B. malayi*, *C. brenneri*, *C. briggsae*, *C. japonica*, *C. remanei*, *O. volvulus*, *P. pacificus* and *S. ratti*.
2. *Caenorhabditis* only - *C. elegans* genes which have at least one orthologues gene in one of the *C. brenneri*, *C. briggsae*, *C. remanei* and *C. japonica* species, and have no orthologues gene in any of the *B. malayi*, *O. volvulus*, *P. pacificus* and *S. ratti* species.
3. Conserved among nematodes - *C. elegans* genes which have at least one orthologues gene in one of the *C. brenneri*, *C. briggsae*, *C. remanei* and *C. japonica* species, and in addition have at least one orthologues gene in one of the *B. malayi*, *O. volvulus*, *P. pacificus* and *S. ratti* species.

## Statistical analysis

For RNAi experiments, Two-way ANOVA tests were used to compare the percentages of the RNAi-affected worms (GFP silencing or fertility for the oma-1 assay) between the tested genotypes. In cases of multiple comparisons between genotypes and across generations, Sidak multiple

comparison tests were applied. For GFP fluorescence experiments, Two-way ANOVA tests were used to compare the normalized GFP expression levels between the genotypes and across the biological repeats. For H3K9me3 qPCR-ChIP experiments Two-way ANOVA tests were used to compare the delta-delta-Ct (or delta-Ct) values between the *gfp* and the *oma-1* loci obtained using two different primer sets. In cases of comparisons between genotypes and loci the Sidak multiple comparison tests were applied. Biological replicates were performed using separate populations of animals. Statistical tests were performed using GraphPad Prism software (Graphpad Prism) version 6. The statistical analysis used for each of the bioinformatics analyses is listed under the corresponding bioinformatics methods.

## Acknowledgements

We thank all the Rechavi lab members for the helpful comments and fruitful discussions. We thank Yael Mor for the fruitful discussions and asistance with formulating the newly evolved genes hypothesis. Some strains were provided by the CGC, which is funded by NIH Office of Research Infrastructure Programs (P40 OD010440). We thank Yosef Shiloh, Yael Ziv, for their assistance and advice. Special thanks to Dror Cohen for the illustrations that he contributed. This work was supported by the ERC (grant #335624) and the Israel Science Foundation (grant #1339/17) and OR gratefully acknowledges the support of the Allen Discovery Center of the Paul G Allen Frontiers Group and the support of the Adelis foundation (no. 01430001000).

## Additional information

### Funding

| Funder | Grant reference number | Author |
|---|---|---|
| Israel Science Foundation | 1339/17 | Itamar Lev<br>Hila Gingold<br>Oded Rechavi |
| European Research Council | 335624 | Itamar Lev<br>Hila Gingold<br>Oded Rechavi |
| Adelis Foundation | 01430001000 | Oded Rechavi |
| Paul G. Allen Family Foundation | | Oded Rechavi |

The funders had no role in study design, data collection, and interpretation, or the decision to submit the work for publication.

### Author contributions

Itamar Lev, Conceptualization, Formal analysis, Investigation, Visualization, Methodology, Writing—original draft, Project administration, Writing—review and editing; Hila Gingold, Conceptualization, Software, Formal analysis, Investigation, Methodology, Writing—original draft, Project administration, Writing—review and editing; Oded Rechavi, Conceptualization, Supervision, Funding acquisition, Writing—original draft, Project administration, Writing—review and editing

### Author ORCIDs

Itamar Lev (iD) http://orcid.org/0000-0002-9100-5100
Oded Rechavi (iD) http://orcid.org/0000-0001-6172-3024

### Decision letter and Author response

Decision letter https://doi.org/10.7554/eLife.40448.028
Author response https://doi.org/10.7554/eLife.40448.029

## Additional files

### Supplementary files

• Supplementary file 1. Sequence of anti-*gfp* RNAi targeting dsRNA used in this study.
DOI: https://doi.org/10.7554/eLife.40448.015

• Supplementary file 2. List of gene targets of endo-siRNAs that depend on SET-25 or SET-32 and their conservation. The table includes all the *C. elegans* genes we examined (20,447 genes). Each gene is marked for being: A. a target of SET-25 dependent endo-siRNAs, B. a target of SET-32 dependent endo-siRNAs. C. Conserved among nematodes, D. Conserved only in *Caenorhabditis*, E. Unique to *C. elegans.*
DOI: https://doi.org/10.7554/eLife.40448.016

• Supplementary file 3. Fold change enrichment values for gene targets of endo-siRNAs that depend on SET-25 or SET-32. The table lists the fold change enrichment values and the matching p-values found in gene targets of SET-25 and SET-32 dependent endo-siRNAs for other genes sets discussed in this manuscript.
DOI: https://doi.org/10.7554/eLife.40448.017

• Transparent reporting form
DOI: https://doi.org/10.7554/eLife.40448.018

### Data availability

All data generated or analyzed during this study are included in the manuscript and supporting files.

The following previously published datasets were used:

| Author(s) | Year | Dataset title | Dataset URL | Database and Identifier |
|---|---|---|---|---|
| Lev I, Seroussi U, Gingold H, Bril R, Anava S, Rechavi O | 2017 | MET-2-Dependent H3K9 Methylation Suppresses Transgenerational Small RNA Inheritance | https://www.ncbi.nlm.nih.gov/geo/query/acc.cgi?acc=GSE94798 | NCBI Gene Expression Omnibus, GSE94798 |
| McMurchy AN, Stempor P, Gaarenstroom T | 2017 | A team of heterochromatin factors collaborates with small RNA pathways to combat repetitive elements and germline stress | https://www.ncbi.nlm.nih.gov/geo/query/acc.cgi?acc=GSE87524 | NCBI Gene Expression Omnibus, GSE87524 |
| Kalinava N, Ni JZ, Gajic Z, Kim M, Ushakov H, Gu SG | 2018 | *C. elegans* Heterochromatin Factor SET-32 Plays an Essential Role in Transgenerational Establishment of Nuclear RNAi-Mediated Epigenetic Silencing. | https://www.ncbi.nlm.nih.gov/geo/query/acc.cgi?acc=GSE117662 | NCBI Gene Expression Omnibus, GSE117662 |
| Klosin A, Casas E, Hidalgo C, Vavouri T | 2017 | Transgenerational transmission of environmental information in *C. elegans* | https://www.ncbi.nlm.nih.gov/geo/query/acc.cgi?acc=GSE83528 | NCBI Gene Expression Omnibus, GSE83528 |

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
