## [Decision Letter]

[**Editorial note:** This article has been through an editorial process in which the authors decide how to respond to the issues raised during peer review. The Reviewing Editor's assessment is that all the issues have been addressed.]

Thank you for submitting your article "H3K9me3 is Required for Inheritance of Small RNAs that Target a Unique Subset of Newly Evolved Genes" for consideration by *eLife*. Your article has been reviewed by four peer reviewers, and the evaluation has been overseen by a Reviewing Editor and Patricia Wittkopp as the Senior Editor. The reviewers have opted to remain anonymous.

The Reviewing Editor has highlighted the concerns that require revision and/or responses, and we have included the separate reviews below for your consideration. If you have any questions, please do not hesitate to contact us.

As you will see in the reviews below, the reviewers agreed on the importance of this topic and the potential impact of the work. However, they also identified concerns with the statistical analyses and strength of the evidence supporting conclusions drawn. During the post-review discussion, after reading the other reviews, at least one of the reviewers stated that s/he became more concerned about the statistics and the reviewer who raised the statistical concerns feels the paper should not be published without quite significant changes. I hope the reviewers' detailed comments below will help you decide how to address these concerns and best proceed with this work. At this point, given the concerns expressed below, it may be best to withdraw this manuscript and return it at a later date when you have had a chance to address the quite significant issues raised by the reviewers.

Separate reviews (please respond to each point):

*Reviewer #1:*

This paper explores interesting observations, that some particular genes respond differently to RNAi from each other. Oma-1 is a germ line gene that for some reason ignores histone methylation pathways whereas GFP does not ignore these pathways. The authors make the interesting connection that GFP is a foreign gene and therefore may be surveilled more than the endogenous *oma-1*. And when they compare siRNAs that are loaded into the germline inheritance pathway, they find that genes that novel to *C. elegans*, more likely to be foreign, are enriched. This is definitely worth publishing. I am not informaticist so I cannot judge the scatterplots in Figures 4,5 so I only weigh in on the RNAi assays for GFP fluorescence or fertility. Those results in Figures 1,2,3 are convincing. I am a little ashamed of myself for not being able to judge the statistics which is why I choose anonymity. Paper is good to me. The discussion is excellent too.

One suggestion: an often overlooked feature of the GFP fusion genes that everyone in *C. elegans* uses is that the introns are artificial and constructed by Andy Fire back when dinosaurs roamed the Earth. If any intron is likely to be viewed as foreign it would be these introns. Deep sequencing of strains carrying GFP fusion genes should show if these introns are highly subject to siRNA production. Need to filter based on the intron exon junctions of these artificial genes. This is not a key suggestion. Optional for the authors.

*Reviewer #2:*

In this paper, the authors attempt to parse the differential roles of the three known *C. elegans* H3K9 histone methyltransferases (HMTs), *set-32, set-25*, and *met-2*. One of the first observations they note is the differential role *set-32* plays in transgenerational epigenetic silencing of the *oma-1* locus and a *gfp* transgene. More specifically, they find that *set-32* is not required to silence *oma-1* but is required to silence *gfp*. This suggests that there may be differential roles between silencing of genes derived endogenously and exogenously. They use ChIP-qPCR to show that this finding is not a result of differential H3K9me3 deposition at the indicated targets. The authors use the minimal data they collected regarding differences in silencing *oma-1* and *gfp* in *set-32* mutants coupled with data from previous studies (predominantly, Kalinava 2017, Towbin 2012, Lev 2017) to draw conclusions about the epistasis of the three HMTs. The focus of their paper seems to pivot from epistasis analysis of *set-32* to the function and targets of *set-25* where they implicate this HMT in specifically targeting newly evolved genes. They identify 151 protein-coding genes that are the specific targets of SET-25 silencing. These genes are enriched for HRDE-1 and WAGO-1 targets, H3K9me3 presence, germline expression, and most interestingly, newly evolved genes. The computational finding that *set-25* specifically regulates newly evolved genes is novel and intriguing, yet, experimentally unvalidated.

The implications of this work would be significant as it represents an important and valuable synthesis of data from multiple labs that have struggled to clearly address to this point, frequently arriving at apparently contradictory results. Though the question is interesting, the authors fail to arrive at a convincing conclusion from their analysis. The experiments and lack rigor and the computational analysis is opaque. The very weak first half of the paper, in which the authors piece together datasets from multiple authors, coupled with the second half of the paper that lacks validated experimental evidence to support their model makes this paper unsuited for publication in *eLife*.

Major Critiques:

1) Figures 1, 2, 3: The first three figures could be combined into one figure. The data are incomplete and, as presented, offer little novel insight. To build a cohesive story, including both novel data and condensation/replication of the numerous published studies referenced in text and outlined below, RNAi inheritance assays must be replicated in all single, double, and triple mutants, minimally including both an exogenous and endogenous target, and results correlated with H3K9me3 deposition at each target in each genetic mutant. This is essential to support the major claims of this paper.

a) Figure 1B (upper and lower) – redundant with Kalinava et al., 2017. (different experimental scheme, same data and interpretation)

b) Figure 1C (lower) – Redundant with Spracklin et al., 2017.

c) Figure 1D – Redundant with Figures 1B-C.

d) Figure 1A-B – Zeller et al., 2016 demonstrate that *met-2;set-25* worms have reduced fertility at 20 and 25 degrees C due to accumulation of DNA damage in the germline. This will impact the total% fertile hermaphrodites used as a readout for anti-*oma-1* RNAi inheritance when analyzing inheritance defects across all HMT mutants.

e) Figure 2A – Loss of H3K9me3 at the *oma-1* locus in treated and untreated worms in the *met-2;set-25;set-32* mutant is redundant with Kalinava et al., 2017. Accumulation of WT levels of H3K9me3 marks across the genome is known to rely on MET-2, SET-25, and SET-32 as shown in Towbin et al., 2012, Mao et al., 2015, and Zeller et al., 2016.

f) Figure 2B – Demonstrates that H3K9me3 signal is higher on the *gfp* locus than *oma-1* locus in untreated worms, yet maintains that since there is no observable fold change between H3K9me3 in treated/untreated (2A) this observation does not account for the difference in RNAi inheritance phenotypes observed in figure 1. This assumption is discrediting a valid interpretation that increased base level H3K9me3 deposition at the *gfp* loci compared to *oma-1* is important for the reported differences in RNAi inheritance.

g) Figure 3 – this figure in insufficient to support the epistasis argument presented in text. It should be condensed into Figure 1. Additionally, the figure contains a typo that renders it uninterpretable: either the data are switched or the y-axis is incorrect.

2) Figure 4: This figure requires more rigorously controlled analysis and should focus on novel insight.

a) Figure 4A – The authors fail to do due diligence in showing that the *set-25* targets they identify are solely *set-25* targets, and that they do not overlap with other HMT targets. Performing *set-32* RNA-seq is a necessary control to ensure that these are not a set of common targets simply hypersensitive to loss of a single HMT.

b) Figure 4B-C – It is unsurprising that targets of an H3K9me3 methyltransferase are enriched for H3K9me3. Additionally, it logically follows that these targets are downstream of a pathway known to deposit H3K9me3 (HRDE/WAGO) and not a licensing pathway (CSR). However, the authors should discuss why these targets appear oocyte specific (shown in 4C), while simultaneously HRDE-1 target-enriched. HRDE targets germline non-expressed genes, pseudogenes, transposons, etc. The authors fail to explain or comment on this paradox.

3) Figure 5: Experimental validation is necessary to uphold the claims of this figure, and the ultimate conclusions of this paper. Failure to experimentally and convincingly show that SET-25 is specifically required to target newly evolved genes renders this paper as nothing more than a prediction founded on unsubstantiated computational outputs lacking critical controls. These controls are especially important in datasets containing small sample sizes (151 identified SET-25 genes) which are more prone to bias.

a) Figure 5A-B – the authors show that SET-25 targets are enriched for newly evolved genes. However, they did not perform the important converse experiment: what percentage of newly evolved genes are SET-25 targets? This analysis is necessary to isolate common identifiers between the subset of newly evolved genes targeted by SET-25 as compared to all newly evolved genes. Without this analysis, their claims into the function of SET-25 are baseless.

b) The computational analysis lacks validation by experimental evidence. A computational model is most useful when it is predictive, and the authors have not demonstrated that here. There are multiple ways in which the authors can test their model, a few of which are listed below:

i) Test multiple transgenes for silencing +/- *set-25*

ii) Validate the transgenerational silencing difference of some of the 151 identified "foreign" SET-25 targets as compared to additional endogenous loci.

iii) Add "foreign" introns or interfere with splice junctions of known (licensed) genes and ChIP-qPCR for H3K9me3 +/- *set-25*

iv) Mutate *gfp* transgene to include introns of highly conserved gene or introduce better splice donors/acceptors

4) Figure 6: The model figure must include the full scope of the work. Currently, it only incorporates data collected in Figures 4-5, but does not help the reader understand the genetic interplay of the HMT pathway, and the relationship between H3K9me3 deposition and RNAi inheritance elucidated throughout the first half of the paper.

Minor Critiques:

1) Figure 1D – these data are redundant to panels (1A-1C). Although the data do illustrate the weak RNAi inheritance responses in later generations, this can still be seen in the previous panels, and the space would be better utilized expanding Figure 1 as outlined in point 1 above. This data could instead be included as a supplemental figure. Furthermore, it is unclear exactly what is being graphed. The Materials and methods section implies they did ImageJ quantification of GFP signal and that they analyzed ~1350 – 1450 germlines, however, there do not appear to be that many points on the graph. Better graph description or clarification of data collection is necessary.

2) Grammar and punctuation throughout – including incorrect possessives (third paragraph of Introduction: "Worms small RNAs…"), missing commas, incomplete sentences (final sentence of Results section titled "Set-25 is required for the maintenance of a specific class of endo-siRNAs"), etc.

3) Figure 3 results typo – "…*met-25;set-25;set-32* triple…" is incorrect (no met-25)

4) Citing primary literature – especially in Introduction. (e.g. First intro sentence should include primary citations in addition to "reviewed in Rechavi and Lev, 2017")

5) Figure 3 requires a schematic of the genetic pathway of the three HMTs. A figure here, or incorporated into the final model, would be beneficial in helping the reader understand the complex genetics at play here.

6) Discussing enrichment of SET-25-dependent endo-siRNA targets – GFP is not a newly-evolved gene. It is, understandably, a proxy for the purposes of this paper, but to define it as such significantly changes the meaning, especially with regards to the definition used throughout Figure 5.

*Reviewer #3:*

Rechavi and colleagues propose that, in *C. elegans*, H3K9me3 is important for siRNA-based silencing of newly evolved genes based on the depletion of siRNAs from a very narrow set of genes, that tend to be *C. elegans*-specific, in an H3K9me3-defective mutant, *set-25*. The data is consistent with a correlation existing between genes that are non-conserved and dependence on *set-25* for siRNA production. It is not clear, however, if this applies specifically to the subset of genes for which siRNAs are dependent on *set-25* or if it is a general theme for all genes targeted by the WAGO/Mutator branch of the siRNA pathway. A revised approach for analyzing the data, described below, would help to clarify this point.

It would be helpful if the Results section included a description of the *gfp* transgene including what endogenous elements it includes and if the endogenous elements are expressed in the germline normally.

Figure 1D. The results and figure legend sections describing Figure 1D would be clearer if there was a brief description of what is being measured.

Figure 2A. Do the open and closed circles represent two different primer pairs as in 2B?

Figure 2B. As this is a qPCR assay, results from two distinct primer sets shouldn't be directly compared without absolute quantification, which is not described as being done in the Materials and methods. It's also a bit concerning that qPCR results from completely distinct experiments – *oma-1* from one experiment and *gfp* from a different experiment done at a different time – are compared side by side.

How many biological replicates were included int the small RNA sequencing experiment in Figure 4A? Even though the sequencing data was described previously, as this is the major data in the manuscript, additional details should be provided – strains, developmental stages, replicates, synchronization, total mapped reads in each library, library preparation method, etc.

What was the fold change cutoff for identifying the 279 genes – 1.2 fold, as was used in describing transposons?

It appears that few genes have more than a very modest reduction or increase in siRNA levels. If the downstream analysis was done with only genes that were depleted of siRNAs by 3x it might lead to more pronounced effects.

Figure 5. A general concern with the data in Figure 5, is that while the results may be significant, and when you have Ns as large as several thousand it doesn't take much difference to be significant, the effects are quite modest and as such the biological relevance is a bit questionable. For example, the difference in PATC density appears to be less than 10% – is there biological relevance to a motif being present 3.5 times compared to 3.8 times? Similarly, do the authors believe that there is biological relevance in a set of genes having a median of 3.6 vs 4 introns?

There must be thousands of *C. elegans*-specific genes, but only a tiny fraction are dependent on *set-25* for optimal siRNA production. Thus, *set-25*/H3K9me3 is presumably a minor player in regulating these so-called newly evolved genes. What is the role of non-*set-25* dependent siRNAs in regulating these genes?

Are the *C. elegans*-specific genes more likely to be targeted by the subset of *set-25* dependent siRNAs than non-*set-25* dependent WAGO/Mutator siRNAs?

In each of the plots in Figure 5, it would be more appropriate if the comparisons of *set-25* targets was to WAGO targets as opposed to all coding genes, as the observed results could potentially apply to any random set of WAGO targets and may not distinguish the subset of genes affected by *set-25*. This is sort of eluded to in the manuscript but not tested.

*Reviewer #4:*

In this manuscript, Lev et al., demonstrate that H3K9me3 is required for inheritance of RNA silencing at some genes but not others. Specifically, H3K9me3 seems to be required for RNA silencing at foreign and newly evolved genes. This is a very intriguing idea, especially because there has been some disagreement as to when and how H3K9me3 is required for RNAi and inheritance.

Overall, this paper makes some interesting observations, especially between *oma-1* and GFP, but this analysis could really be taken to the next level by more broadly by addressing how consistent this observation is genome-wide (and beyond just the *set-25* mutant small RNA data). Also, the discovery of SET-25 targeting recently evolved genes is not surprising because of the significant overlap with ERGO-1, WAGO-1, and HRDE-1 target genes. Multiple publications have shown that these ERGO/WAGO/HRDE targets are overall less conserved, have fewer introns, and are spliced less efficiently, so SET-25-dependent siRNA target genes also having some of these features seems expected.

Specific suggestions

Need more/better background about the three methyltransferases. What about tissue-specific expression? Presumably all three are expressed in the germline? A chart or table summarizing the inheritance phenotypes (previously published and new) associated with each methyltransferase alone and in combination with each other would also be useful, either in Figure 1 or the supplement.

Figure 1.

While Figure 1 does a nice job summarizing the differences in inheritance between GFP and *oma-1*, much of this data is redundant with Kalinava, 2017, and Spraklin, 2017. To make this point more conclusively, inheritance assays of several more target genes (endogenous vs. exogenous) would be necessary to demonstrate that this is really a difference between these two types of genes, rather than something specific to *oma-1* or GFP uniquely.

Figure 2.

In B, H3K9me3 levels should be measured across more genes. Do SET-25 dependent siRNA target genes have H3K9me3 at levels similar to GFP? What about siRNA target genes that are not SET-25 dependent? Is there an overall pattern? This could also be analyzed more thoroughly with the McMurchy data set.

Figure 4. In B, how are random set controls generated? Should be indicated briefly in figure legend and in Materials and methods

Figure 5A. While PATC, rnp-2, and # of introns are all technically statistically significant, the graphs are not especially convincing. The differences are quite subtle and the majority of the SET-25 genes overlap with the bulk of the protein coding genes. Indicating the mean numerically in each panel (for PATC density, fold enrichment, or # introns) could help.

How is the bootstrap control performed for panel B? No indication in the Materials and methods.

Bigger picture questions/suggestions –

Do endogenous *set-25*-dependent siRNA target genes also lose siRNAs in a heritable way? Could be addressed through crosses between *set-25* and wild-type, followed by sequencing or rt-qPCR.

There is somewhat of a disconnect between the first few figures, looking at the relationship between the three methyltransferases, and the later figures looking only at SET-25-dependent siRNA targets. How do the SET-25-dependent siRNA target genes compare to the small RNAs dependent on the other to H3K9 methyltransferases (MET-2 and SET-32, alone or in combination)? Is it possible that other HRDE/WAGO/ERGO target genes (or specifically other newly evolved genes) are methylated by other methyltransferases?

It would be relatively easy to look at H3K9me3 across all targets of the HRDE/WAGO/ERGO pathways similarly to Figure 4B. How does the H3K9me3 signal change in the methyltransferase single, double and triple mutants?

Does loss of H3K9me3 at endogenous targets change the mRNA expression of these genes?

Minor Comments:

Figure 1. The diagram in panel A could be more clear. I generally assume a lightening bolt is indicated mutagenesis, not RNAi. Need to make it clear that RNAi is only occurring in generation 1 and subsequent generations are moved to OP50 plates.

Need to indicate in the results and figure legend what GFP transgene is being used for this assay.

How is GFP + or – defined. Materials and methods describe a calculation for fluorescence intensity but figure legend indicates% worms with GFP fluorescence. Is there a cutoff above which GFP is "on" and below which GFP is "off"?

As for the aesthetics of Figure 1, in panel D, wild-type should either be above the mutants, or ideally, they should in the same graph for easier comparison. It is a little bit confusing as to which labels go with which graphs and which wild-type data goes with which mutant data.

Labeling in Figure 2 is also confusing. Open vs. closed circles should be indicated on graph. Colors should be consistent between A and B.

Figure 3 should be moved to be part of Figure 1.

Figure 4. Are tissue enrichment boxes in C supposed to be colored? Text says fold enrichment for oocytes is 1.24 – shouldn't this be colored pale pink?

Figure 5. Is the number of stars for P value cutoffs consistent between different panels of this figure? Panel B is also made difficult to interpret because one's eye is drawn to blue dots, rather than the red bars. D and E, could be zoomed in on the lower part of the graph, to potentially allow for observation of a difference in the means (especially in D, which is supposedly different from the control).

Typos in text:

Subsection “SET-25-dependent endo-siRNAs target a unique subset of newly evolved genes”, first paragraph – "SET-25-dependnt endo-siRNAs"

Last sentence of results ends in comma.

Double period at the end of Figure 5 legend.

Figure 3—figure supplement 1A "Radom set control"

---

## [Author Response]

Reviewer #1:

This paper explores interesting observations, that some particular genes respond differently to RNAi from each other. Oma-1 is a germ line gene that for some reason ignores histone methylation pathways whereas GFP does not ignore these pathways. The authors make the interesting connection that GFP is a foreign gene and therefore may be surveilled more than the endogenous oma-1. And when they compare siRNAs that are loaded into the germline inhertance pathway, they find that genes that novel to C. elegans, more likely to be foreign, are enriched. This is definitely worth publishing. I am not informaticist so I cannot judge the scatterplots in Figures 4,5 so I only weigh in on the RNAi assays for GFP fluorescence or fertility. Those results in Figures 1,2,3 are convincing. I am a little ashamed of myself for not being able to judge the statistics which is why I choose anonymity. Paper is good to me. The discussion is excellent too.

We thank the reviewer for the kind words.

One suggestion: an often overlooked feature of the GFP fusion genes that everyone in C. elegans uses is that the introns are artificial and constructed by Andy Fire back when dinosaurs roamed the Earth. If any intron is likely to be viewed as foreign it would be these introns. Deep sequencing of strains carrying GFP fusion genes should show if these introns are highly subject to siRNA production. Need to filter based on the intron exon junctions of these artificial genes. This is not a key suggestion. Optional for the authors.

That is a great suggestion, but endo-siRNAs target exons (almost exclusively), as RdRP amplification it thought to occur on the spliced mRNA (Sapetschnig et al., 2015). Nevertheless, we examined the siRNAs that are aligned to the *gfp* construct that we used in our study, and our analysis (Figure 4—figure supplement 2) shows no targeting of the synthetic introns of the GFP construct. Therefore this does not explain how *gfp* is different from *oma-1*.

Reviewer #2:

In this paper, the authors attempt to parse the differential roles of the three known C. elegans H3K9 histone methyltransferases (HMTs), set-32, set-25, and met-2. One of the first observations they note is the differential role set-32 plays in transgenerational epigenetic silencing of the oma-1 locus and a gfp transgene. More specifically, they find that set-32 is not required to silence oma-1 but is required to silence gfp. This suggests that there may be differential roles between silencing of genes derived endogenously and exogenously. They use ChIP-qPCR to show that this finding is not a result of differential H3K9me3 deposition at the indicated targets. The authors use the minimal data they collected regarding differences in silencing oma-1 and gfp in set-32 mutants coupled with data from previous studies (predominantly, Kalinava 2017, Towbin 2012, Lev 2017) to draw conclusions about the epistasis of the three HMTs. The focus of their paper seems to pivot from epistasis analysis of set-32 to the function and targets of set-25 where they implicate this HMT in specifically targeting newly evolved genes. They identify 151 protein-coding genes that are the specific targets of SET-25 silencing. These genes are enriched for HRDE-1 and WAGO-1 targets, H3K9me3 presence, germline expression, and most interestingly, newly evolved genes. The computational finding that set-25 specifically regulates newly evolved genes is novel and intriguing, yet, experimentally unvalidated.The implications of this work would be significant as it represents an important and valuable synthesis of data from multiple labs that have struggled to clearly address to this point, frequently arriving at apparently contradictory results.

We thank the reviewer and agree that lots of data were collected over the years, and that it is still a challenge to connect all the dots.

Though the question is interesting, the authors fail to arrive at a convincing conclusion from their analysis. The experiments and lack rigor and the computational analysis is opaque. The very weak first half of the paper, in which the authors piece together datasets from multiple authors, coupled with the second half of the paper that lacks validated experimental evidence to support their model makes this paper unsuited for publication in eLife.

We think, and the other 3 reviewers appear to share our view, that our conclusions stand, and that this critique is not justified. We cannot refute this statement (since no specific problems are mentioned, it’s just general negativity), and therefore we choose instead to address all the specific comments, one-by-one.

Major Critiques:1) Figures 1, 2, 3: The first three figures could be combined into one figure. The data are incomplete and, as presented, offer little novel insight.

The three figures show different things, we think lumping all these data together would be too overwhelming. Therefore, for the sake of clarity, we leave the figures separated. However, in line with this suggestion, reviewers #3 and #4 suggested that we move Figure 3 to the supplementary material, and we did so in the revised manuscript.

To build a cohesive story, including both novel data and condensation/replication of the numerous published studies referenced in text and outlined below, RNAi inheritance assays must be replicated in all single, double, and triple mutants, minimally including both an exogenous and endogenous target, and results correlated with H3K9me3 deposition at each target in each genetic mutant. This is essential to support the major claims of this paper.

Indeed it’s always important to replicate previous results, and we made sure to do it. We explicitly say when we replicate previous findings, for example see Results section “We successfully replicated the results of Kalinava et al., and came to the same conclusion ….”, and subsection “The levels of RNAi-induced H3K9me3 do not explain the gene-specific requirements of methyltransferases for heritable RNAi”: “we found, as expected (Kalinava et al., 2017)”. Now rephrased following this comment: " Using qPCR we found, as was discovered before "

We do not think every single experiment from the past must be repeated (specifically not all the experiments that we did ourselves in the past), this would not add new information and no genetic backgrounds are missing from our current analysis, that could change any conclusion. Please see the new Figure 1—figure supplement 2, where we summarize all the conclusions that were made in the past and all the new conclusions, with regard to the H3K9 methylation mutants and their effect on RNAi inheritance.

a) Figure 1B (upper and lower) – redundant with Kalinava et al., 2017. (different experimental scheme, same data and interpretation)

As the reviewer says: It’s a different experimental scheme, it’s not the same experiments as conducted by Kalinava et al. (they examined mRNA expression by real time PCR, and we examine the phenotype). Additionally, Kalinava et al. did not analyze anti-GFP RNAi, and obviously the comparison with GFP is the main point here (in the figure). Therefore this figure is not redundant.

In the previous comment the reviewer asked that we replicate all previous findings, and here he/she says we shouldn’t, so the different concerns contradict.

b) Figure 1C (lower) – Redundant with Spracklin et al., 2017.

Figure 1C is not redundant since Spracklin et al., 2017 did not analyze heritable silencing of *oma-1*. In contrast to heritable silencing of *gfp* (which was studied by Sparcklin et al), and we show that heritable silencing of *oma-1* does not require *set-32*.

c) Figure 1D – Redundant with Figures 1B-C.

Figures 1D (Now Figure 1—figure supplement 1) shows the absolute quantification of GFP in contrast to the binary quantification (expressing / not expressing) presented in Figures 1B and C. We think this is helpful information and therefore do not find this figure redundant. The figure shows, moreover, that there’s weak inheritance in the methyltransferase mutants, which is different from what was previously suggested in the literature (Sparkling et al., 2017).

d) Figure 1A-B – Zeller et al., 2016 demonstrate that met-2;set-25 worms have reduced fertility at 20 and 25 degrees C due to accumulation of DNA damage in the germline. This will impact the total% fertile hermaphrodites used as a readout for anti-oma-1 RNAi inheritance when analyzing inheritance defects across all HMT mutants.

This difference can’t explain our conclusion since in the *met-2, set-25 and set-32* mutants we do not see a reduction, on the contrary, we see an *increase* (due to silencing of *oma-1)* in the number of live (hatching) progeny. Therefore, if anything, when we say RNAi inheritance is still functional in these mutants we *underestimate* the effect.

e) Figure 2A – Loss of H3K9me3 at the oma-1 locus in treated and untreated worms in the met-2;set-25;set-32 mutant is redundant with Kalinava et al., 2017.

We explicitly say in the text that we replicated Kalinava et al’s results. As the reviewer noted himself/herself above, replication is valuable. In the text, we wrote: “we found, as expected (Kalinava et al., 2017)”. Now rephrased following this comment: "Using qPCR we found, as was discovered before."

Moreover, this result was necessary for the comparisons of RNAi-induced H3K9me3 levels deposited on the *gfp* and *oma-*1 genes.

Accumulation of WT levels of H3K9me3 marks across the genome is known to rely on MET-2, SET-25, and SET-32 as shown in Towbin et al., 2012, Mao et al., 2015, and Zeller et al., 2016.

We use the reduction in RNAi-induced H3K9me3 levels as a control. We did not claim that this is one of the novelties of our paper. (And in any case Towbin et al. and Zeller et al. did not examine RNAi-induced H3K9me3, and we have previously shown that this is different than non RNAi-induced K9me.).

f) Figure 2B – Demonstrates that H3K9me3 signal is higher on the gfp locus than oma-1 locus in untreated worms, yet maintains that since there is no observable fold change between H3K9me3 in treated/untreated (2A) this observation does not account for the difference in RNAi inheritance phenotypes observed in Figure 1. This assumption is discrediting a valid interpretation that increased base level H3K9me3 deposition at the gfp loci compared to oma-1 is important for the reported differences in RNAi inheritance.

The reviewer misunderstood, we did say that the higher base levels of H3K9me3 on gfp loci compared to *oma-1* could be important for the reported differences in RNAi inheritance, please see subsection “Endo-siRNAs that depend on H3K9me3 methyltransferases target a distinctive subset of newly evolved genes”: “Further, we find that newly evolved genes have higher levels of H3K9me3.., Likewise, in the absence of RNAi, in wild-type animals, gfp, the newly evolved gene that we investigated, has higher levels of H3K9me3, in comparison to the well-conserved *oma-1* gene“. To further clarify this in the revised manuscript we now refer to this conclusion again earlier in the paper, in subsection “The levels of RNAi-induced H3K9me3 do not explain the gene-specific requirements of methyltransferases for heritable RNAi”.

g) Figure 3 – this figure in insufficient to support the epistasis argument presented in text.

We do not agree, and moreover this observation is in accordance with new studies that show that SET-32 plays a role in the initial steps of RNAi inheritance (Kalinava et al., 2018). We added this statement to the revised manuscript in subsection “SET-32 acts upstream to MET-2 and SET-25 to support RNAi inheritance”.

It should be condensed into Figure 1.

This figure was moved to the supplementary information, now Figure 1—figure supplement 2.

Additionally, the figure contains a typo that renders it uninterpretable: either the data are switched or the y-axis is incorrect.

Thank you, the typo is now corrected.

2) Figure 4: This figure requires more rigorously controlled analysis and should focus on novel insight.

We do not agree and do not know how this critique could be used to improve the paper, as it’s not clear what would satisfy the reviewer’s definition of “novel insight” (and he/she does not specify which controls are missing). In contrast, we think this is a very useful and novel figure. To the revised figure we also added new data on *set-32* (see below) and WAGO targets that further strengthen our conclusions.

a) Figure 4A – The authors fail to do due diligence in showing that the set-25 targets they identify are solely set-25 targets, and that they do not overlap with other HMT targets.

We never claimed that we saw anything that is true in general to all known HMTs. The Gu lab recently sequenced small RNAs from *set-32* mutants, the data was released (December 2019) 5 months after our manuscript was submitted (so obviously we couldn’t analyze it before, and therefore there was no “due diligence” to make). In the revised manuscript we include an analysis of these newly released data (Figures 3 and 4 and their figure supplements), which further support our conclusions and show that SET-25-dependent endo-siRNAs targets are unique yet their characteristics are shared with SET-32-dependent endo-siRNAs targets (for example high H3K9me3, newly evolved genes, HRDE-1 targets..).

Performing set-32 RNA-seq is a necessary control to ensure that these are not a set of common targets simply hypersensitive to loss of a single HMT.

An analysis of *set-32* RNA-seq is added to the revised manuscript, see Figures 3 and 4. We found that endo-siRNAs that depend on the two HMTs (SET-32 and SET-25) target different genes (very small overlap of 25 out of 465 genes). Importantly, however, the characteristics of these gene targets are very similar – H3K9me3 methylated, known targets of HRDE-1 and ERGO-1 argonauts, and newly evolved.

b) Figure 4B-C – It is unsurprising that targets of an H3K9me3 methyltransferase are enriched for H3K9me3.

We do not agree. It is certainly surprising that endogenous small RNAs are *directly* affected by H3K9 methylation. The effects of the methylations could have been indirect (as is the case sometimes, for example as we demonstrated in Lev et al. Current Biology 2017). Further, in general it’s not a valid critique to speculate if something is surprising or not.

Additionally, it logically follows that these targets are downstream of a pathway known to deposit H3K9me3 (HRDE/WAGO) and not a licensing pathway (CSR). However, the authors should discuss why these targets appear oocyte specific (shown in 4C), while simultaneously HRDE-1 target-enriched. HRDE targets germline non-expressed genes, pseudogenes, transposons, etc. The authors fail to explain or comment on this paradox.

We don’t think it’s a paradox. HRDE-1 is expressed specifically in the germline. Gene targets of HRDE-1 are not completely silenced in the germline. Indeed, we find a large overlap between HRDE-1 targets (Buckley et al., 2012) and germline expressed genes ((Buckley et al., 2012), (1000/1048 HRDE-1 targets!), See Figure 3—figure supplement 2. We elaborate on this now in subsection “H3K9me3 methyltransferases are required for the biogenesis of a specific class of endo-siRNAs”.

3) Figure 5: Experimental validation is necessary to uphold the claims of this figure, and the ultimate conclusions of this paper. Failure to experimentally and convincingly show that SET-25 is specifically required to target newly evolved genes renders this paper as nothing more than a prediction founded on unsubstantiated computational outputs lacking critical controls. These controls are especially important in datasets containing small sample sizes (151 identified SET-25 genes) which are more prone to bias.

It's not clear how to test this (that newly evolved genes are subjected to SET-25-dependent heritable RNAi). The data generated an interesting hypothesis that maybe future studies could further study. In addition, the experiments with the two reporter genes (*oma-1* and *gfp*) support these ideas.

a) Figure 5A-B – the authors show that SET-25 targets are enriched for newly evolved genes. However, they did not perform the important converse experiment: what percentage of newly evolved genes are SET-25 targets? This analysis is necessary to isolate common identifiers between the subset of newly evolved genes targeted by SET-25 as compared to all newly evolved genes. Without this analysis, their claims into the function of SET-25 are baseless.

The converse fold change of enrichment of SET-25 targets amongst newly evolved genes is 2.57 (the same as the direct fold change enrichment value presented in the paper) and the is p-value < 0.0001.

The fold enrichment is calculated (described in the Materials and methods section) as follows:

Enrichment=ObservedExpected=fractionofgenesbelongtothei-thsetamongdifferentialexpressedSTGsfractionofgenesbelongtothei-thsetamongallthegenes

Therefore, converse enrichment calculations do not give values different from the enrichment values. The P value might change due to the randomization and changes in the size of the examined gene set.

b) The computational analysis lacks validation by experimental evidence. A computational model is most useful when it is predictive, and the authors have not demonstrated that here. There are multiple ways in which the authors can test their model, a few of which are listed below:i) Test multiple transgenes for silencing +/- set-25

We previously tested an additional transgene for silencing with *set-25* mutants and got the same results (see Sup Figure 2B in Lev et al., 2017). Others have tested different *gfp* transgenes and reached similar conclusions (Ashe et al., 2012; Spracklin et al., 2017).

ii) Validate the transgenerational silencing difference of some of the 151 identified "foreign" SET-25 targets as compared to additional endogenous loci.

We add new mRNA analyses that show that some of these genes are re-expressed in *set-25* mutants (subsection “H3K9me3 methyltransferases are required for the biogenesis of a specific class of endo-siRNAs”).

iii) Add "foreign" introns or interfere with splice junctions of known (licensed) genes and ChIP-qPCR for H3K9me3 +/- set-25iv) Mutate gfp transgene to include introns of highly conserved gene or introduce better splice donors/acceptors

These experiments are extremely time consuming and very technically challenging, and therefore are out of the scope of the paper (this could be a stand-alone paper). We generated multiple interesting hypotheses that would generate future work.

4) Figure 6: The model figure must include the full scope of the work. Currently, it only incorporates data collected in Figures 4-5, but does not help the reader understand the genetic interplay of the HMT pathway, and the relationship between H3K9me3 deposition and RNAi inheritance elucidated throughout the first half of the paper.

We thank the reviewer for this helpful suggestion. Since we added to the revised manuscript genome wide analyses of *set-32* mutants, we incorporate the conclusion of these experiments to the Model Figure. In addition, we added a supplementary figure that explains the interplay between the HMT mutants (Figure 1—figure supplement 2).

Minor Critiques:1) Figure 1D – these data are redundant to panels (1A-1C). Although the data do illustrate the weak RNAi inheritance responses in later generations, this can still be seen in the previous panels, and the space would be better utilized expanding figure 1 as outlined in point 1 above. This data could instead be included as a supplemental figure.

As we wrote above (this was raised in a previous comment of the same reviewer) Figure 1D (Figure 1—figure supplement 1 in the revised version) shows the absolute quantification of GFP, and therefore we do not think it is redundant. Moreover, it shows that there’s weak inheritance in the mutants (which is different from what was previously suggested in the literature). Nevertheless, we have no objection to moving it to the supplemental information, and we did so in the revised manuscript (now shown in Figure 1—figure supplement 1).

Furthermore, it is unclear exactly what is being graphed. The Materials and methods section implies they did ImageJ quantification of GFP signal and that they analyzed ~1350 – 1450 germlines, however, there do not appear to be that many points on the graph. Better graph description or clarification of data collection is necessary.

Since it’s a scatter column graph, points with the similar values “fall” on each other. We clarify this better in the legend of the revised version.

2) Grammar and punctuation throughout – including incorrect possessives (third paragraph of Introduction: "Worms small RNAs…"), missing commas, incomplete sentences (final sentence of Results section titled "Set-25 is required for the maintenance of a specific class of endo-siRNAs"), etc.

We corrected every typo that we could detect.

3) Figure 3 results typo – "…met-25;set-25;set-32 triple…" is incorrect (no met-25)

Fixed.

4) Citing primary literature – especially in Introduction. (e.g. First intro sentence should include primary citations in addition to "reviewed in Rechavi and Lev, 2017")

We don’t think this is a pattern in the paper, and believe in certain cases (where too much work must be referred to) it’s appropriate to reference reviews, but in this specific case we added additional references to primary papers.

5) Figure 3 requires a schematic of the genetic pathway of the three HMTs. A figure here, or incorporated into the final model, would be beneficial in helping the reader understand the complex genetics at play here.

We now added a summarizing supplemental figure that gives more background on the three different methyltransferase. See Figure 1—figure supplement 2B.

6) Discussing enrichment of SET-25-dependent endo-siRNA targets – GFP is not a newly-evolved gene. It is, understandably, a proxy for the purposes of this paper, but to define it as such significantly changes the meaning, especially with regards to the definition used throughout Figure 5.

We agree, indeed, we made sure that we did not say it’s a newly evolved gene anywhere in the paper. As the reviewer writes, we say that it could perhaps be seen as a proxy. We toned down these statements in the revised text.

Reviewer #3:

Rechavi and colleagues propose that, in C. elegans, H3K9me3 is important for siRNA-based silencing of newly evolved genes based on the depletion of siRNAs from a very narrow set of genes, that tend to be C. elegans-specific, in an H3K9me3-defective mutant, set-25. The data is consistent with a correlation existing between genes that are non-conserved and dependence on set-25 for siRNA production. It is not clear, however, if this applies specifically to the subset of genes for which siRNAs are dependent on set-25 or if it is a general theme for all genes targeted by the WAGO/Mutator branch of the siRNA pathway. A revised approach for analyzing the data, described below, would help to clarify this point.

We thank the reviewer for his/her excellent comments, we added the suggested new analyses to the revised manuscript.

It would be helpful if the Results section included a description of the gfp transgene including what endogenous elements it includes and if the endogenous elements are expressed in the germline normally.

Added (Figure 4—figure supplement 2E).

Figure 1D. The results and figure legend sections describing Figure 1D would be clearer if there was a brief description of what is being measured.

Added to the figure legend. (Now Figure 1—figure supplement 1).

Figure 2A. Do the open and closed circles represent two different primer pairs as in 2B?

Yes, we wrote it in the legend but further clarified in the revised text since it obviously wasn’t clear enough.

Figure 2B. As this is a qPCR assay, results from two distinct primer sets shouldn't be directly compared without absolute quantification, which is not described as being done in the Materials and methods. It's also a bit concerning that qPCR results from completely distinct experiments – oma-1 from one experiment and gfp from a different experiment done at a different time – are compared side by side.

The qPCR results shown in Figure 2B are normalized to a common control gene *eft-*3, as noted in the figure legends. In the revised manuscript we show data from the *gfp* experiment of an additional germline-expressed gene, *dpy-*28, that shows that this gene has low levels of H3K9me3, similarly to the *oma-1* gene, and lower than *gfp* (Figure 2B). It is also worth noting that the *gfp* and *oma-*1 H3K9me3 ChIP experiments were done using the same strain (SX1263), grown on the same control bacteria, using the same ChIP protocol and reagents.

How many biological replicates were included int the small RNA sequencing experiment in Figure 4A? Even though the sequencing data was described previously, as this is the major data in the manuscript, additional details should be provided – strains, developmental stages, replicates, synchronization, total mapped reads in each library, library preparation method, etc.

We had 3 biological replicates (explained in the statistical analysis part of the Materials and methods). All the other details that the reviewer asked for are also added to the revised manuscript. See Information on the sequencing libraries analyzed in this paper section in the Materials and methods.

What was the fold change cutoff for identifying the 279 genes – 1.2 fold, as was used in describing transposons?It appears that few genes have more than a very modest reduction or increase in siRNA levels. If the downstream analysis was done with only genes that were depleted of siRNAs by 3x it might lead to more pronounced effects.

In this analysis we did not use a fold change cutoff, but an adjusted p-value cutoff of the DESeq2 algorithm. Adding a fold change cutoff did not change our results (Supplementary file 3).

Figure 5. A general concern with the data in Figure 5, is that while the results may be significant, and when you have Ns as large as several thousand it doesn't take much difference to be significant, the effects are quite modest and as such the biological relevance is a bit questionable. For example, the difference in PATC density appears to be less than 10% – is there biological relevance to a motif being present 3.5 times compared to 3.8 times? Similarly, do the authors believe that there is biological relevance in a set of genes having a median of 3.6 vs 4 introns?

That’s a good point, we agree with the reviewer, and therefore we acknowledge this in the revised manuscript and detail the exact differences in means. It also worth noting that the PATC density distributions were graphed in log2 scale such that the actual differences are larger than seen (~10% in median, and ~25% in average levels). In the revised version we note in the figure that log2 scale is shown in addition to the explanation in the figure legends. We also added analysis of PATC density levels of targets of WAGO associated endo-siRNAs (now Figure 4A) and endo-siRNAs *upreulgated* in *set-25* and *set-32* mutants (Figure 4—figure supplement 1) that shows that the reduced PATC density levels are specific to endo-siRNAs that depend on SET-25 or SET-32. Moreover, in the revised paper we further explain how many complementary signals, even if some are relatively small (in average intron number or PATC levels) could together expose genes that need to be silenced (See subsection “Endo-siRNAs that depend on H3K9me3 methyltransferases target a distinctive subset of newly evolved genes” paragraph three).

There must be thousands of C. elegans-specific genes, but only a tiny fraction are dependent on set-25 for optimal siRNA production. Thus, set-25/H3K9me3 is presumably a minor player in regulating these so-called newly evolved genes. What is the role of non-set-25 dependent siRNAs in regulating these genes?Are the C. elegans-specific genes more likely to be targeted by the subset of set-25 dependent siRNAs than non-set-25 dependent WAGO/Mutator siRNAs?

Another good point. We analyzed this in Figure 4—figure supplement 3 in the revised manuscript. We found that the ERGO-1, WAGO-1, and HRDE-1 and Mutator class small RNAs show significant enrichment for targeting newly evolved genes. However, while the fold enrichment for newly evolved genes is comparable between endo-siRNAs that depend on SET-25 or SET-32, and those that depend on other WAGO/Mutator factors, we found that the SET-25/32 dependent endo-siRNAs that target newly evolved genes do not fully overlap with the endo-siRNAs that depend on the WAGO pathways. Further, the enrichment for newly evolved genes is maintained for SET-25/32 dependent endo-siRNAs, even when we removed the targets shared with ERGO-1, WAGO-1, and HRDE-1 and Mutator from the analysis.

In each of the plots in Figure 5, it would be more appropriate if the comparisons of set-25 targets was to WAGO targets as opposed to all coding genes, as the observed results could potentially apply to any random set of WAGO targets and may not distinguish the subset of genes affected by set-25. This is sort of eluded to in the manuscript but not tested.

These comparisons are added to the revised manuscript. See Figures 3 and 4.

Reviewer #4:

[…] Specific suggestions –Need more/better background about the three methyltransferases. What about tissue-specific expression? Presumably all three are expressed in the germline? A chart or table summarizing the inheritance phenotypes (previously published and new) associated with each methyltransferase alone and in combination with each other would also be useful, either in Figure 1 or the supplement.

Excellent idea, we now added a summarizing supplement figure that gives more background on the three different methyltransferase. See Figure 1—figure supplement 2.

Figure 1.While Figure 1 does a nice job summarizing the differences in inheritance between GFP and oma-1, much of this data is redundant with Kalinava, 2017, and Spraklin, 2017. To make this point more conclusively, inheritance assays of several more target genes (endogenous vs. exogenous) would be necessary to demonstrate that this is really a difference between these two types of genes, rather than something specific to oma-1 or GFP uniquely.

Unfortunately there are no good RNAi inheritance assays for other endogenous germline-expressed genes (a problem the field deals with for a long while). The *oma-1* assay that Alcazar et al. developed is special, perhaps because they found a redundant dominant lethal and temp sensitive germline allele that is perfect for RNAi inheritance assays.

In response to this comment we add new data (Figure 1—figure supplement 3) showing that we tried to develop new assays for examination of heritable silencing of other endogenous germline genes, *fog-2* and *sup-35* (since these have phenotypes that can be followed across generations). Unfortunately, we find that silencing of these genes is not inherited transgenerationally, and therefore these genes cannot be used as tools.

Figure 2. In B, H3K9me3 levels should be measured across more genes. Do SET-25 dependent siRNA target genes have H3K9me3 at levels similar to GFP? What about siRNA target genes that are not SET-25 dependent? Is there an overall pattern? This could also be analyzed more thoroughly with the McMurchy data set.

We added this analysis to the revised manuscript, see Figure 4. Our analysis of H3K9me3 levels on WAGO-associated endo-siRNA targets, which is based on the McMurchy data, shows that these targets are methylated to a lesser degree in comparison with genes targets of SET-25-dependent- and SET-32- dependent endo-siRNAs.

Figure 4. In B, how are random set controls generated? Should be indicated briefly in figure legend and in Materials and methods

This was written in the Materials and methods, we added this now to the legends as well.

Figure 5A. While PATC, rnp-2, and # of introns are all technically statistically significant, the graphs are not especially convincing. The differences are quite subtle and the majority of the SET-25 genes overlap with the bulk of the protein coding genes. Indicating the mean numerically in each panel (for PATC density, fold enrichment, or # introns) could help.How is the bootstrap control performed for panel B? No indication in the Materials and methods.

We added the mean to the graph panels and text. It is also important to note that the PATC data is presented in log2 scale such that the actual differences are larger than what they may seem in the graph (~10% in median and ~25% on average), and that such reduction in PATC density was not found in the new controls we added (Figure 4A and Figure 4—figure supplement 1). We also explain now in the text how the cumulative effect of small differences in intron numbers and PATC might allow recognition of genes that need to be silenced (See subsection “Endo-siRNAs that depend on H3K9me3 methyltransferases target a distinctive subset of newly evolved genes” paragraph four). We also explain how the bootstrap control is performed (previously it was in the Materials and methods, now we add it also to the legends).

Bigger picture questions/suggestions –Do endogenous set-25-dependent siRNA target genes also lose siRNAs in a heritable way? Could be addressed through crosses between set-25 and wild-type, followed by sequencing or rt-qPCR.

We think these small RNAs are heritable since they are bound by Argonauts involved in small RNA inheritance such as HRDE-1 and WAGO-1. This could be interesting to further study in the future.

There is somewhat of a disconnect between the first few figures, looking at the relationship between the three methyltransferases, and the later figures looking only at SET-25-dependent siRNA targets. How do the SET-25-dependent siRNA target genes compare to the small RNAs dependent on the other to H3K9 methyltransferases (MET-2 and SET-32, alone or in combination)? Is it possible that other HRDE/WAGO/ERGO target genes (or specifically other newly evolved genes) are methylated by other methyltransferases?

We added to the revised manuscript an analysis of small RNAs from *set-32* mutants, and we compare it to *set-25* mutants (Figures 3 and 4, and their figure supplements, subsection “H3K9me3 methyltransferases are required for the biogenesis of a specific class of endo-siRNAs”). We described the changes in small RNAs in *met-2* mutants in Lev et al. Current Biology 2017, and therefore did not go into it again, also because we think (as we described in that paper), that the effects of MET-2 are often indirect.

It would be relatively easy to look at H3K9me3 across all targets of the HRDE/WAGO/ERGO pathways similarly to Figure 4B.

We added these analyses to the revised text, see Figure 4. It shows that these WAGO targets are marked by H3K9me3, but targets of SET-25 or SET-32 dependent endo-siRNAs have higher levels of H3K9me3.

How does the H3K9me3 signal change in the methyltransferase single, double and triple mutants?

By analyzing the recent Kalinava data set (Kalinava et al., 2018), we tested how the H3K9me3 levels on SET-32-dependent endo-siRNA target genes change in *set-32* mutants and found, as might be expected, that gene targets of SET-32-dependent endo-siRNAs have reduced levels of H3K9me3 in *set-32* mutants (Figure 3—figure supplement 1, discussed in paragraph two of subsection “H3K9me3 methyltransferases are required for the biogenesis of a specific class of endo-siRNAs”).

Does loss of H3K9me3 at endogenous targets change the mRNA expression of these genes?

We added this analysis to the revise text. We find that targets of SET-25-dependent endo-siRNAs are significantly enriched with genes that are re-expressed in *set-25* mutants (Klosin et al., 2017) see paragraph two of subsection “H3K9me3 methyltransferases are required for the biogenesis of a specific class of endo-siRNAs”.

Minor Comments:Figure 1. The diagram in panel A could be more clear. I generally assume a lightening bolt is indicated mutagenesis, not RNAi. Need to make it clear that RNAi is only occurring in generation 1 and subsequent generations are moved to OP50 plates.Need to indicate in the results and figure legend what GFP transgene is being used for this assay.How is GFP + or – defined. Materials and methods describe a calculation for fluorescence intensity but figure legend indicates% worms with GFP fluorescence. Is there a cutoff above which GFP is "on" and below which GFP is "off"?

Thank you, all these clarifications were made in the Figure, figure legends and Materials and methods.

As for the aesthetics of Figure 1, in panel D, wild-type should either be above the mutants, or ideally, they should in the same graph for easier comparison. It is a little bit confusing as to which labels go with which graphs and which wild-type data goes with which mutant data.

We now place them one above the other in Figure 1—figure supplement 1.

Labeling in Figure 2 is also confusing. Open vs. closed circles should be indicated on graph. Colors should be consistent between A and B.

Fixed.

Figure 3 should be moved to be part of Figure 1.

We moved this figure to the Supporting materials, we think it was too overwhelming. Now Figure 1—figure supplement 2.

Figure 4. Are tissue enrichment boxes in C supposed to be colored? Text says fold enrichment for oocytes is 1.24 – shouldn't this be colored pale pink?

Fixed.

Figure 5. Is the number of stars for P value cutoffs consistent between different panels of this figure?

Definitely, yes.

Panel B is also made difficult to interpret because one's eye is drawn to blue dots, rather than the red bars. D and E, could be zoomed in on the lower part of the graph, to potentially allow for observation of a difference in the means (especially in D, which is supposedly different from the control).

Fixed. We removed the dots, and with regards to Figure D, we show the median values in the figures.

Typos in text:Subsection “SET-25-dependent endo-siRNAs target a unique subset of newly evolved genes”, first paragraph – "SET-25-dependnt endo-siRNAs"Last sentence of results ends in comma.Double period at the end of Figure 5 legend.Figure 3—figure supplement 1A "Radom set control"

Thank you. All fixed.